. Pathogens

# Engineering a multivalent antibody nanoparticle to overcome SARS-CoV-2 Omicron immune evasion

Hui Sun[1,2¤◉], Yanan Jiang[1,2◉], Miaolin Lan[1,2◉], Ming Zhou[1,2◉], Gangshun Yi[3,4], Juan Shen[3], Tingting Deng[1,2], Liqin Liu[1,2], Yang Huang[1,2], Yu Li[1,2], Jinfu Su[1,2], Yanling Lin[1,2], Zhenqin Chen[1,2], Lizhi Zhou[1,2], Tingting Li[1,2], Hai Yu[1,2], Tong Cheng[1,2], Yali Zhang[1,2], Lunzhi Yuan[1,2], Shaowei Li[1,2]*, Ying Gu[1,2]*, Peijun Zhang[3,4], Ningshao Xia[1,2]*, Qingbing Zheng[1,2]*

1 State Key Laboratory of Vaccines for Infectious Diseases, Xiang An Biomedicine Laboratory, Department of Laboratory Medicine, School of Public Health, School of Life Sciences, Xiamen University, Xiamen, China, 2 National Institute of Diagnostics and Vaccine Development in Infectious Diseases, State Key Laboratory of Molecular Vaccinology and Molecular Diagnostics, The Research Unit of Frontier Technology of Structural Vaccinology of Chinese Academy of Medical Sciences, Xiamen University, Xiamen, China, 3 Division of Structural Biology, Centre for Human Genetics, University of Oxford, Oxford, United Kingdom, 4 Diamond Light Source, Harwell Science and Innovation Campus, Didcot, United Kingdom

◉ These authors contributed equally to this work.
¤ Current address: Tianfu Jincheng Laboratory, Chengdu, China
* shaowei@xmu.edu.cn (SL); guying@xmu.edu.cn (YG); nsxia@xmu.edu.cn (NX); abing0811@xmu.edu.cn (QZ)

## Abstract

The rapid evolution of SARS-CoV-2 and the subsequent emergence of Omicron subvariants pose significant challenges to the efficacy of existing vaccines and therapeutics, including those previously reported most broad neutralizing antibodies (bnAbs). Here, we investigated the molecular basis of the altered neutralization profile of a bnAb, 1C4, against recent variants. 1C4 is effective against early variants from Alpha to Omicron BQ.1, but is circumvented by BQ.1.1, XBB and thereafter variants, primarily due to an additional R346T mutation that diminishes its binding affinity. Cryo-electron microscopy analysis revealed that despite the loss of neutralizing potency, 1C4 retained residual binding to the spike protein of immune-evasive variants such as XBB, which harbor altered receptor-binding domain (RBD). Furthermore, 1C4 exhibited a diminished capacity to inhibit ACE2 engagement with Omicron variants, amplifying the intricacies of viral immune evasion tactics. To address this, we employed the mi3-SpyCatcher-based nanoparticle to polymerize 1C4 (mi3-1C4), which reestablished the neutralization potency against recent variants by enhancing avidity via multivalent binding. Such multivalent binding can promote efficient spike aggregation as well as viral cross-linking, thereby providing enhanced protection against both the infection of Beta and XBB variants in a hamster model. Together, our findings delineate the molecular landscape of immune evasion by neutralizing

**Data availability statement:** Structure coordinates have been deposited in the Protein Data Bank under accession codes 9W14 (WT-S:1C4-interface), 9KVJ (BA.5-S:1C4-interface), 9KVK (WT-S:8H12:1C4:3E2-interface), 9KVQ (BA.1-S:8H12:1C4:3E2-interface), 9KVT (BA.2-S:8H12:1C4:3E2-interface), 9KWY (WT-RBD:1C4:ACE2) . The corresponding EM density maps have been deposited in the Electron Microscopy Data Bank under accession numbers EMD-65523 (WT-S:1C4), EMD-60852 (BA.5-S:1C4-state 1), EMD-60862 (BA.5-S:1C4-state 2), EMD-62596 (BA.5-S:1C4-interface), EMD-62597 (WT-S:8H12:1C4:3E2-interface), EMD-60863 (WT-S:8H12:1C4:3E2), EMD-608864 (BA.1-S:8H12:1C4:3E2), EMD-62599 (BA.1-S:8H12:1C4:3E2-interface), EMD-60974 (BA.2-S:8H12:1C4:3E2), EMD-62597 (BA.2-S:8H12:1C4:3E2-interface), EMD-60975 (BA.2.75-S:8H12:1C4:3E2) and EMD-60976 (BA.5-S:8H12:1C4:3E2-state 1), EMD-60979 (BA.5-S:8H12:1C4:3E2-state 2), EMD-60963 (XBB-S:8H12:1C4:3E2),EMD-62601 (WT-RBD:1C4:ACE2) and EMD- 65522 (mi3-1C4).

**Funding:** This work was supported by grants from the National Natural Science Foundation of China (32170942 to Q.Z and 81991491 to N.X), the Wellcome Investigator Award (206422/Z/17/Z), and the European Research Council AdG grant (101021133) to P.Z. The funders had no role in study design, data collection and analysis, decision to publish, or preparation of the manuscript. None of the authors received salary support from the funding agencies mentioned.

**Competing interests:** The authors have declared that no competing interests exist.

antibodies and provide strategic insight for the adaptation of antibody engineering to keep pace with viral evolution.

## Author summary

As SARS-CoV-2 evolves, new variants evade immunity, reducing the efficacy of previously potent antibodies like 1C4. We identified key mutations in the viral spike protein responsible for its loss of neutralization and developed a nanoparticle-based strategy to restore its activity. By enhancing antibody potency through multi-valent binding, our approach effectively neutralizes resistant variants and improves protection in animal models. This strategy offers a promising solution to counteract viral immune escape and extend the durability of antibody therapeutics, addressing the challenge of keeping pace with rapidly mutating viruses.

## Introduction

The ongoing coronavirus disease 2019 (COVID-19) pandemic caused by Severe Acute Respiratory Syndrome Coronavirus 2 (SARS-CoV-2) has left an indelible mark, with over 770 million confirmed cases and over 6.9 million fatalities globally [1]. Despite the widespread vaccination, the emerged Omicron sublineages, such as XBB.1.5 [2], EG.5 [3,4], BA.2.86 [5] and JN.1 subvariants [6], have raised concerns about the efficacy of existing vaccines and therapeutics. Previous clinical data have shown that the boost of the WT/BA.5 bivalent vaccine fails to produce potent neutralization against subsequent subvariants [7], indicative of antibody-mediated immune escape by the newly emerged variants. Notably, JN.1 and its derivatives, which exhibit heightened immune evasion, have spread extensively worldwide throughout 2024 and continue to mutate, perpetuating their impact [6,8].

The COVID-19 pandemic has accelerated the development of a wide range of antibody therapies designed to prevent or treat infection.The receptor binding domain (RBD) of the spike (S) protein is the principal target for the neutralizing antibody (nAb) response against SARS-CoV-2. With the ongoing evolution of the Omicron sublineages, which display pronounced antigenic shift, specific mutation hotspots such as R346, K444, V445, G446, and N460, particularly in subvariants such as BQ.1.1, XBB, and EG.5, have had a marked impact on the efficacy of vaccines and therapeutic antibodies [9–11]. Neutralizing antibodies, derived from either post-infection or vaccine-elicited serum, are mainly categorized into five distinct classes (Classes 1–5) [12–14], many of which are prone to immune evasion [10]. For instance, Class 1/2 nAbs are notably susceptible to mutations present in early Omicron variants [15], such as E484A and Q493R, and their neutralizing capacity is further reduced by the F486 substitution found in the BA.4/5 subvariant [16]. Meanwhile, Classes 3 and 4 nAbs, which target relatively conserved neutralizing epitopes, are sensitive to mutations R346T, V445P, D405N, and R408S [10, 17]. Even for S309, a widely utilized

broad-spectrum nAb authorized for emergency COVID-19 treatment, has experienced notable declines in potency due to detrimental mutations such as S371F, G339D/H, and R346T [11,18,19]. The attenuation of S309's neutralizing capacity is particularly pronounced against the newly BA.2.86 variant [5]. This suggests the efficacy of most existing antibodies might eventually be entirely negated in the future, emphasizing the formidable task of developing antibodies that match the rapid viral evolution.

Innovative strategies, such as bispecific antibody engineering [20,21], nanoparticle-based antibody platforms [22–24], and modular nanobody platform based on IgM scaffold [25], have been explored to augment the performance of existing antibodies. These modifications have been shown to preserve the neutralizing breadth against prospective variants, thereby alleviating the challenges of tailoring antibodies to rapidly evolving variants within limited timeframes. Such advancements have demonstrated the capacity to enhance antibody avidity, conferring increased neutralization potency against a spectrum of variants. However, the detailed molecular mechanisms underlying these enhancements remain incompletely understood, underscoring the need for further structural and functional investigations to optimize their design and application.

In this study, we conducted a biochemical and structural characterization of a Class 3 bnAb which could neutralize BQ.1 along with its antecedent subvariants, yet is evaded by BQ.1.1 and XBB variants, primarily due to an additional R346T mutation that diminishes its binding affinity. Crucially, our systematic investigation of immune complexes formed by wild-type (WT) and various Omicron subvariants spike proteins revealed a reduced, yet detectable, binding capacity of the bnAb 1C4 to a range of Omicron spikes. Nevertheless, the increased mutational burden within the 1C4 epitopes results in an attenuated affinity of nAbs along with a compromised ability to block ACE2, synergistically promotes viral immune evasion. To address the reduced affinity of 1C4 toward XBB, we developed an innovative mi3-SpyCatcher nanoparticle-based strategy for antibody polymerization. This strategy proved effective in enhancing the binding affinity of the polymerized 1C4 (mi3-1C4), thereby reestablishing its neutralization potency against the newly emerged XBB and EG.5 variants. mi3-1C4's multivalent binding facilitated efficient spike aggregation of Omicron spikes including JN.1, as well as native SARS-CoV-2 virus cross-linking. Moreover, mi3-1C4 provided robust protection against XBB infection in hamster model. Our detailed insights into the mechanism of antibody evasion and the strategy of neutralization restoration offer a valuable framework for the strategic development of antibodies tailored to counter future variants of SARS-CoV-2.

## Results

### Binding and neutralizing characteristics of nAb 1C4 against SARS-CoV-2 variants

In our previous study, a SARS-CoV-2 nAb 1C4 was generated using hybridoma technology and identified as a Class 3 nAb [20]. In this study, we conducted a detailed assessment of 1C4's binding affinities to the RBDs of various SARS-CoV-2 variants, including wild-type, Alpha, Beta, Delta, and several Omicron subvariants, using surface plasmon resonance (SPR). The results revealed that 1C4 bound to the WT RBD with high affinity (KD=0.488 nM) and retained comparable affinities to early VOCs such as Alpha, Beta, and Delta (S1 Fig). However, its binding proficiency showed a moderate decline for most Omicron subvariants, including BA.1, BA.2, BA.2.75 and BA.5, owing to their faster dissociation rate, even though it still retained nanomolar affinities ranging from 2.170 nM to 4.890 nM (S1 Fig). This finding indicated that 1C4 recognizes a relatively conservative epitope. Regrettably, 1C4 displayed markedly reduced binding for the XBB RBD with an affinity of merely 8,750 nM (S1 Fig), which is three and near three orders of magnitude lower than that for the WT and BA.5 RBDs, respectively. The reduced binding affinity of 1C4 to the XBB RBD is likely due to the additional mutations at R346 and V445P, which have been shown to compromise the efficacy of many Class 3 nAbs, as previously described [9,10].

To investigate the neutralizing breadth of 1C4, we next evaluated its neutralizing potency against pseudotyped SARS-CoV-2 D614G strain and representative variants including Omicron sublineages BA.1, BA.2, BA.2.75, BA.5, BQ.1, BQ.1.1 and XBB using a lentiviral-based pseudoviruses system [26]. 1C4 showed potent neutralizing activity to SARS-Cov-2

D614G strain with an IC50 value of 32 ng/mL (Figs 1A and S2). Interestingly, 1C4 exhibited a stepwise declined neutralization against subsequent variants from Alpha to Omicron BQ.1, corresponding with 1.2~28.6-fold decrease in IC50 values compared with the SARS-CoV-2 D614G strain (Fig 1B). As for Omicron BQ.1.1 and XBB sublineages, 1C4 was entirely ineffective in neutralization (Fig 1B), although we could still detect out a lower binding affinity of 1C4 to the XBB RBD (KD value, 8,750 nM). These data suggested that the additional K444T mutation present in the BQ.1 RBD when compared to BA.5 may not lead to significant reduction in neutralization efficacy. However, the newly added shared mutation R346T on the BQ.1.1 and XBB RBDs may pose a great threat to the neutralization breadth of 1C4. Accordingly, 1C4 effectively neutralizes SARS-CoV-2 and variants including BQ.1 and pre-BQ.1 variants, while evade by BQ.1.1 and XBB variants.

**Cryo-EM structure of 1C4 complexed with the BA.5 spike reveals its relatively conserved epitope**

To elucidate the epitope characteristics and understand the molecular basis for 1C4's neutralization breadth, we employed cryo-EM single particle analysis to determine the structure of 1C4 Fab complexed with the BA.5 spike trimer (BA.5-S:1C4). The heterogenous refinement and final refinements generated two distinct binding states (hereafter referred to state 1 and 2, state 1: 2-"up" RBD and state 2: 1-"up" RBD) at global nominal resolutions of 3.45 Å and 3.68 Å, respectively (Figs 1C, 1D, S3 and S1 Table). Both states present the three RBDs fully bound with three 1C4 Fabs, demonstrating a recognition of at least one opened RBD by 1C4 to facilitate its binding orientation without strict conformational preference. Given the dynamic feature of the RBD, the interface between 1C4 and the RBD in the global refinement exhibited poor resolution. As such, we conducted local refinement focused on the interface of a single "down" RBD and 1C4-Fab variable regions from the state 1 complex, which resulted in an improved structure at 3.82 Å resolution (S3 Fig), allowing us to build atomic model of the 1C4 variable regions and the RBD (BA.5-RBD:1C4) (Fig 1E).

The atomic model of BA.5-RBD:1C4 revealed that 1C4 binds an epitope surrounding the N343-glycan (Fig 1E). This binding site allows N343-glycan to interact extensively with both the heavy and light chains of 1C4 (Figs 1E and S4), a characteristic reminiscent of the representative nAb S309 and CV38–142 [27,28]. The importance of this glycan-mediated interaction is underscored by the considerable reduction in binding ability of 1C4 to the RBD upon glycan removal using PNGase F (S5 Fig), whereas the removal of this glycan induces only a minor decrease in the binding affinity to S309. 1C4 mainly uses HCDR2, HCDR3 and LCDR3 to contact RBD residues, the hydrophobic patch at the interface between the heavy and light chains of 1C4 facilitates RBD binding (S4A Fig). Meanwhile, the epitope of 1C4 is composed of 10 residues on the RBD including E340, N343, T345, R346, K440, L441, S443, V445, N450 and T500, covering a buried surface area of 865.9 Å$^2$ (Figs 1F and S6). Among these residues, N343, T345, R346 and S443, form an interaction network of six hydrogen bonds with the complementarity-determining regions (CDRs) of 1C4 (Fig 1G).

In contrast to the WT-RBD, the structure of BA.5-RBD:1C4 reveals that the N440K mutation still allows residue 440 to engage in the interaction with 1C4. However, as N/K440 is located at the periphery of the epitope (Fig 1F) and is not involved in the network of hydrogen bonds and salt bridges illustrated in Fig 1G, this mutation does not exert a negative impact on 1C4 binding. However, another critical residue, R346, emerges in the spike proteins of the BQ.1.1 and XBB subvariants. It not only mediates hydrogen bonding with 1C4 but also contributes to a cation-π interaction with W102 of 1C4 heavy-chain (S4B Fig). Undoubtedly, this hotspot residue plays a pivotal role in maintaining stable antibody–antigen interactions. Therefore, the R346T mutation is expected to significantly impair the binding affinity between 1C4 and the mutated spike proteins. We next performed structural modeling to disclose the molecular basis of 1C4 evasion by XBB. The R346 residue involved in a large number of contact interaction, including hydrogen bond interaction, with 1C4 (Fig 1I). Theoretically, the R346T substitution in the XBB RBD leads to a reduction in surface hydrophilicity, thereby disrupting the interactions with residues Y50$^L$ and W102$^H$ of 1C4 (Fig 1I). The reduced side chain length of T346 results in the loss of hydrogen bond and van der Waals interactions with 1C4. Furthermore, the V445 on BA.5 spike interacts with T30$^H$ and Y54$^H$, and it seems that the substitution of V445P on the XBB RBD may also weaken the contacts with Y54$^H$ and the

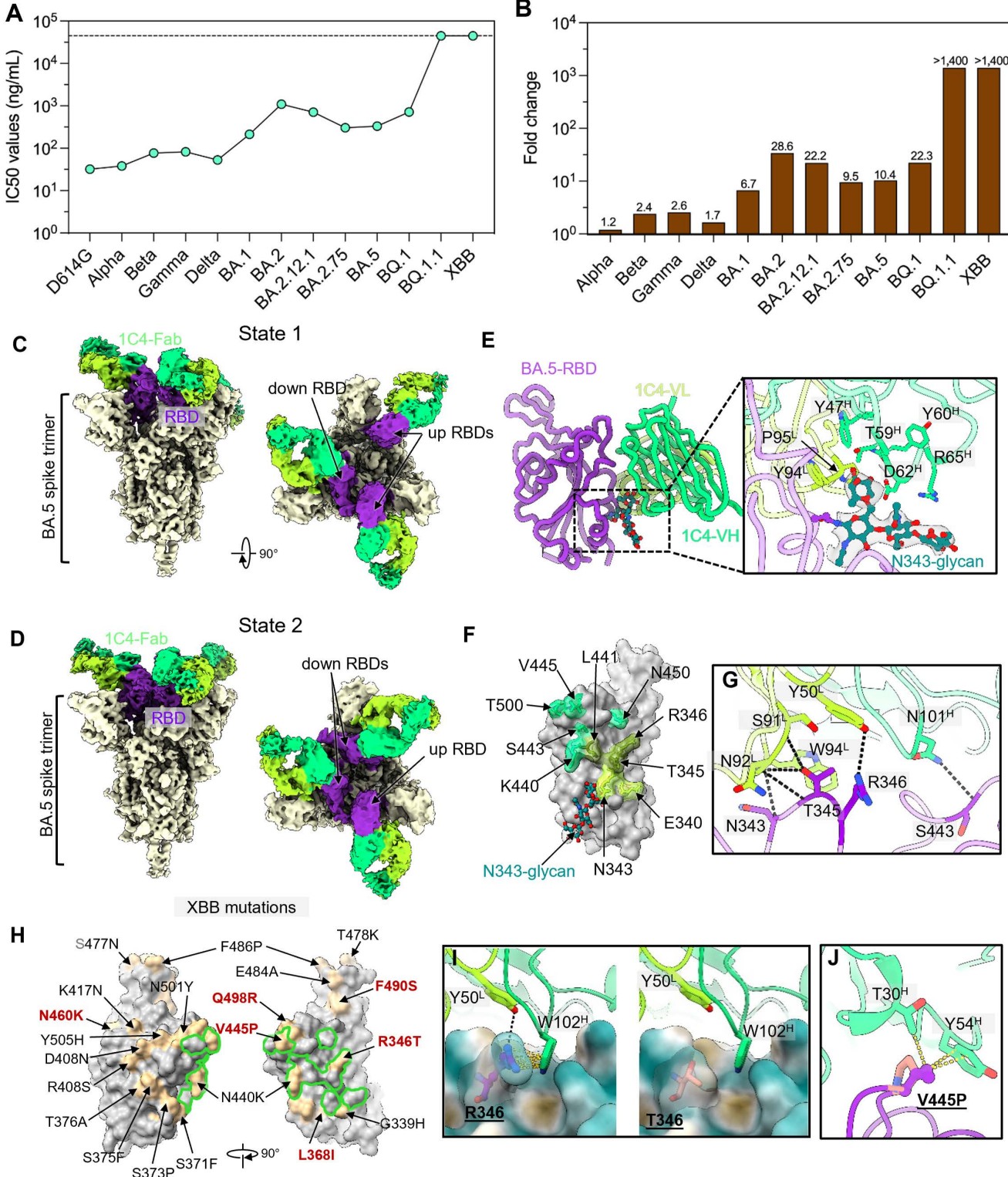

**Fig 1. The neutralizing potency and structural characterization of 1C4. (A)** Comparison of neutralizing potency of 1C4 against SARS-CoV-2 D614G and variants. The dots are shown as IC50 values. Data were collected from three technical replicates and displayed as means±SD. **(B)** Fold change in the neutralization efficacy of 1C4 against SARS-CoV-2 variants as compared with the D614G stain. **(C and D)** The domain-colored cryo-EM

density maps of BA.5-S:1C4 reveals two states of immune complexes structures. State 1 (C) shows one spike bound by three 1C4 Fabs, with two up-conformation RBDs and one down-conformation RBD. State 2 (D) displays that the spike trimer presents two down RBDs and one up RBD, all bound by 1C4 Fabs. **(E)** The cartoon representation of BA.5-RBD:1C4. The N343-glycan is highlighted by sticks. The right panel illustrates the close-up view that shows the interaction details between N343-glycan and 1C4. **(F)** The footprint of 1C4 is mapped on the surface representation of the BA.5 RBD. Residues involved in interactions are shown as stick with a transparent surface representation. The contact region of the heavy chain, light chain and both chains on the RBD are colored in lime, green yellow and olive, respectively. **(G)** Interaction details of the BA.5 RBD bound to 1C4. Residues that participate in hydrogen bonds are shown as stick. The hydrogen bonds are highlighted as dash lines. **(H)** Comparison of the 1C4 footprint and mutations emerging in the XBB subvariant. Each mutation is labeled, with those unique to XBB, relative to BA.5, highlighted in red. **(I and J)** The key mutation residues R346T (I) and V445P (J) on the XBB subvariant disrupt the binding of 1C4.

mutated proline would alter the local bond conformation (Fig 1J). These structural insights indicated that R346 plays a critical role in the binding of 1C4.

### Cryo-EM reveals 1C4's binding capacities to diverse Omicron spikes including XBB

Considering the limited neutralizing breadth of 1C4, we next sought to explore the molecular mechanism underlying the evasion of 1C4 that prevents it from neutralizing the XBB sublineages. In addition to 1C4, we included two structurally characterized antibodies, 8H12 and 3E2, representing non-overlapping epitopes with distinct neutralization profiles (Fig 2A–2C) [20], in order to compare how different epitopes respond to viral evolution. To this end, we prepared immune complexes of the three antibodies with spike proteins from WT and various Omicron subvariants, including BA.1, BA.2, BA.2.75, BA.5, and XBB, enabling a direct visualization of their binding modes at the molecular level. Among them, 8H12 and 3E2 were characterized in our previous study, where 8H12 was shown to neutralize early Omicron variants but was evaded by BA.4/5, while 3E2 displayed reduced neutralization beginning with BA.2 and later subvariants. To directly visualize their binding to diverse spikes, we assembled trimeric spike complexes with all three Fabs. Initially, we determined the WT spike complex (WT-S:1C4:3E2:8H12) at a global resolution of 3.76 Å, and obtained a locally refined structure of the RBD bound by all three Fabs at 3.44 Å (Figs 2D, S7 and S1 Table). The WT spike adopted three up conformation RBDs, each simultaneously decorated with three Fabs (Figs 2D, 2E and S7). A similar binding pattern was observed in the BA.1 spike complex, which exhibited only slightly reduced affinities for the nAbs compared to the WT spike (Figs 2F and S7). Contrastingly, for the BA.2 and BA.2.75 spike complexes, one RBD retained trivalent Fab binding, while the other two were bound only by 1C4 and 8H12, with 3E2 failing to engage (Figs 2F, S7 and S8). Interestingly, we captured and refined the binding interface between 3E2 and the BA.2 RBD, confirming a reduced contact area compared to that of the WT-RBD and BA.1-RBD (Figs 2B and S9). This confirmed that mutations in the BA.2 spike indeed affect 3E2's binding, leading to a significant reduction in 3E2 affinity. In the case of the BA.5 subvariants, although they evade neutralization by 8H12 and 3E2, the complex structure revealed that one of the three RBDs remains decorated by all three Fabs (Fig 2F). Moreover, 1C4 showed relatively stronger Fab density on the BA.5 spike than 8H12 and 3E2, which may reflect differences in binding strength.

Most strikingly, even for the highly evasive XBB spike, structural analysis revealed that a subset of RBDs retained low-level engagement by all three antibodies. Specifically, seven Fabs—three 1C4 Fabs, two 3E2 Fabs, and two 8H12 Fabs—were found bind to the three RBDs on the XBB spike, while only one RBD presented relatively higher-quality densities of all three decorated Fabs, consistent with the low but detectable binding affinity of 1C4 to the XBB RBD. These findings demonstrate that, despite the overall reduction in antibody affinity for the Omicron spikes due to evolution and the accumulating mutations on the RBDs, a fundamental level of stable interaction between nAbs and the spikes still persists. These structural findings, therefore, go beyond visual confirmation of binding. They provide molecular-level evidence that many Omicron subvariants retain low-level antibody engagement, as reflected by the reduced but still detectable number of Fabs bound per spike. Although such engagement is insufficient for neutralization, it may nevertheless provide a structural scaffold for future engineering strategies.

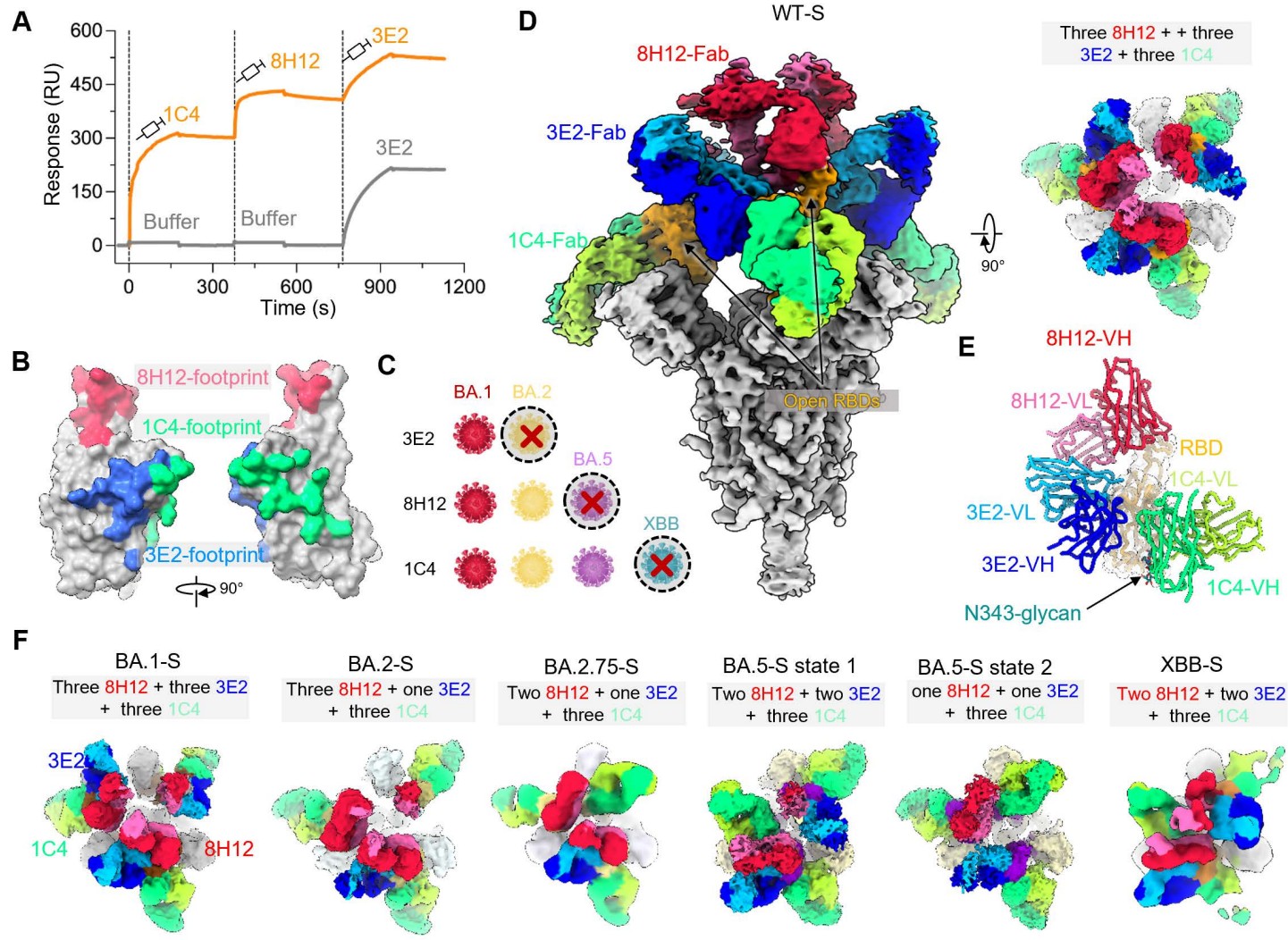

**Fig 2. Cryo-EM structures of 1C4-based triple-nAb complexed with diverse spikes of variants reveal the mechanism of antibody evasion. (A)** The sensorgram of SPR-based competitive assay involving antibodies 8H12, 3E2 and 1C4. Wild-type spike protein (WT-S) was immobilized on a CM5 sensor chip, and antibodies were injected as the analyte in the flow phase. **(B)** Comparison of the epitopes recognized by 8H12, 3E2 and 1C4 on the WT-RBD. **(C)** Schematic diagram of the neutralization potecies of antibodies 8H12, 3E2, and 1C4 against the representative Omicron variants. The red cross (×) indicates that the antibody is unable to neutralize this variant or any of the subsequent Omicron subvariants. The virus cartoon models were created using the paint tool SAI software. **(D)** The domain-colored map of WT-S:1C4:8H12:3E2, each RBD is bound by all three Fab and the spike is bound by nine Fabs in total. **(E)** The atomic model of WT-RBD:1C4:8H12:3E2 with the variable regions of three Fabs. **(F)** Comparison of the cryo-EM structures of the immune complexes of triple-nAb (8H12, 3E2 and 1C4) with the spike proteins of WT, BA.1, BA.2, BA.2.75, BA.5 and XBB, which demonstrates the diverse binding modes and diverse amount of Fabs onto different spikes.

## Accumulation of mutations on the RBD abolishes the ACE2-blocking capacity of 1C4

The above structural information indicates that 1C4 remains binding to all tested spikes of Omicron subvariants. Further ELISA results confirm that 1C4 can bind to all tested spikes, whereas 8H12 and 3E2 completely lose their ability to bind to the BA.5 spike (Fig 3A). To understand the underlying reasons for 1C4's inability to effectively neutralize the XBB variant, we examined its capacity to inhibit ACE2 interaction. All three mAbs could fully block the binding of human ACE2 to the WT spike, with $IC_{50}$ values of 0.174 µg/mL (8H12), 0.270 µg/mL (1C4) and 0.610 µg/mL (3E2), respectively (Fig 3B).

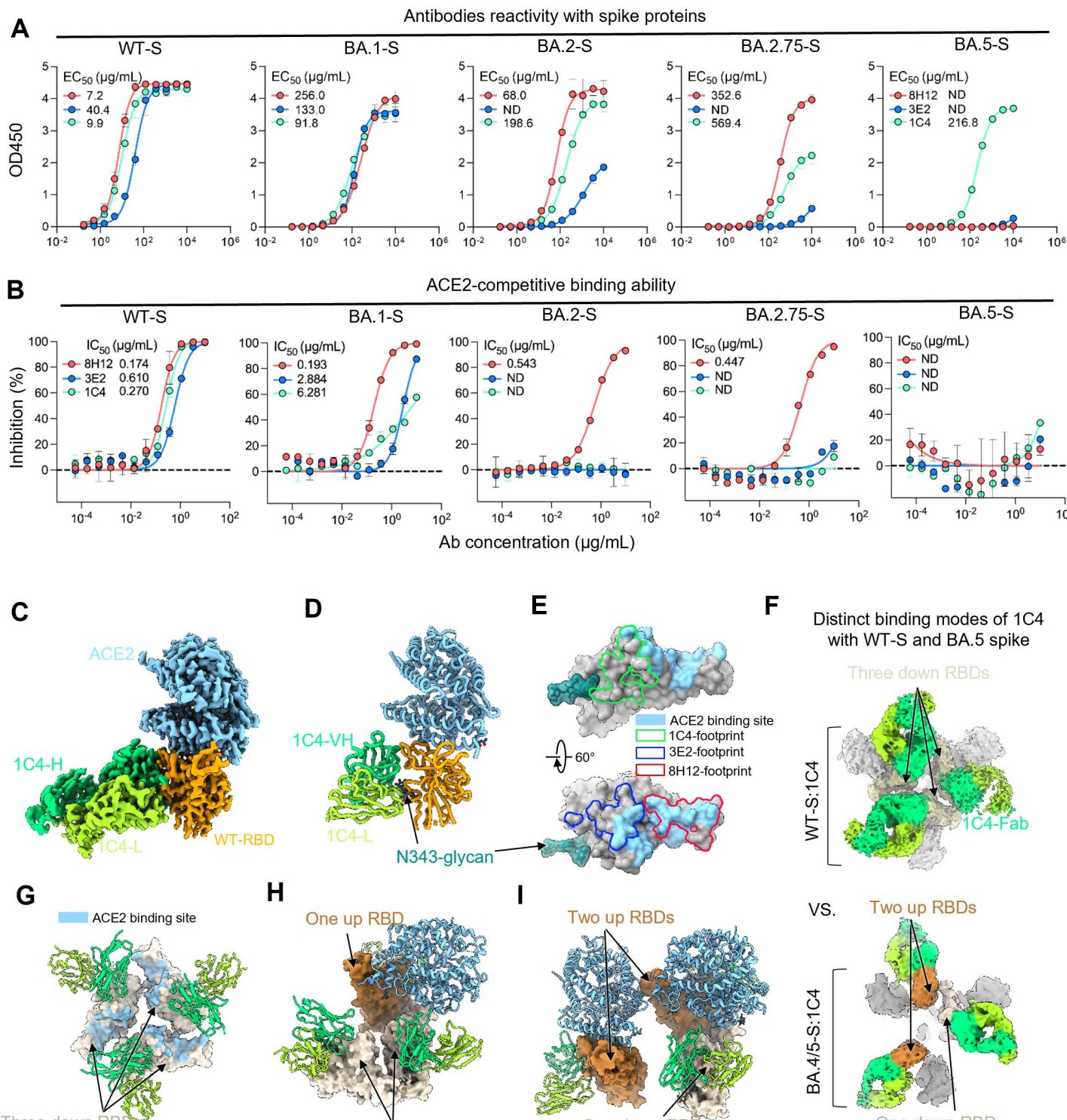

**Fig 3. The molecular basis of the ACE2 blocking diversity of 1C4 against different spikes. (A)** ELISA of antibodies (8H12, 3E2 and 1C4) reactivity with different spike proteins (WT, BA.1, BA.2, BA.2.75 and BA.5). **(B)** ELISA-based identification of 1C4 and two other nAbs for their competition with ACE2 binding on the various spike proteins (WT, BA.1, BA.2, BA.2.75 and BA.5). **(C and D)** Cryo-EM structure of WT-RBD:1C4:ACE2. The domain-colored density map (C) and the corresponding cartoon representation (D) are shown. **(E)** Comparison of the binding footprints of three antibodies as

well as the ACE2. **(F)** The distinct binding modes of 1C4 to the WT- and BA.5-spike revealed by cryo-EM analysis. **(G-I)** Comparison of the ACE2-binding ability for the RBD in WT- and BA.5-spike complexed with 1C4. Three models are generated by fitting the structures onto the corresponding density maps of WT-S:1C4 and BA.5-S:1C4. For the WT-S:1C4, all three RBDs are locked at the down conformation (G) and cannot be bound by the ACE2. While for BA.5-S:1C4, two states of the bound spikes with one (H) or two (I) up RBDs. In either case, the up RBDs can be bound by the ACE2.

However, interestingly, as the RBD mutations accumulated in subsequent variants, all three nAbs progressively lost their ability to block ACE2 binding (Fig 3B). Specifically, 3E2 and 1C4 lost the ACE2-blocking capacity when binding to the BA.2 spike while all three nAbs lost the ACE2 block capacity when binding to the BA.5 spike (Fig 3B). These results confirmed that the ACE2 receptor, with its high affinity to bind to diverse spike proteins, can effectively engage the RBD when nAbs exhibit reduced binding affinities to the mutated spikes of variants.

S309 and several other Class 3 nAbs, were previously identified as incapable of competing with ACE2 for the RBD binding [17,27]. Of particular note, however, 1C4 showed unexpectedly effective blocking potency against ACE2 binding to the WT spike (Fig 3B), contradicting the experiment where ACE2 and 1C4 were bound to the RBD simultaneously (S10 Fig). To further validate the receptor-blocking and neutralizing mechanism of 1C4, we prepared the complex of the WT RBD incubated with 1C4 Fab and ACE2 simultaneously, and successfully determined a 2.82 Å structure of such triple-complex through cryo-EM analysis (Figs 3C, 3D, S11 and S1 Table). The WT-RBD:1C4:ACE2 structure confirmed that 1C4 is not capable to prevent the binding of ACE2 to the WT RBD. The binding epitope of 1C4 does not overlap with that of ACE2, differ from the ACE2-competitive nAbs 8H12 and 3E2 (Fig 3C–3E).

In order to unveil the incongruence between ACE2 blockage assay and structural analysis, we conducted cryo-EM analysis of the WT spike incubated solely with 1C4 Fab. Unexpectedly, the 1C4–WT spike complex exhibits two distinct conformations: one with a two-down, one-up RBD arrangement, and the other with all three RBDs locked in the down state (S11 Fig). As a result, this distinct 'locking' conformation results in all three ACE2-binding sites are rendered inaccessible (Fig 3F), and therefore delineates an unconventional pattern of ACE2 inhibition, similar to a previous study [29]. By contrast, the cryo-EM structure of BA.5-S:1C4 exhibited that 1C4 loses the ability to lock all RBDs in their down states, probably because the BA.5 spike adopts a more open conformation that facilitates RBD exposure [30]. In this case, the complexes harbor at least one "up" RBD, potentially permitting ACE2 binding without steric hindrance (Fig 3H and 3I). This implied the reduced capability of 1C4 to block ACE2, which may correlate with the diminished neutralization against BA.5. Therefore, the conformational selectivity of the RBD-specific antibody may also play a significant role in its neutralizing potency. Additionally, despite detectable binding between 1C4 and the XBB spike, neutralization is completely abrogated, likely due to the R346T mutation significantly reducing 1C4's affinity. Given the nanomolar-level affinity of ACE2 to the Omicron spikes, the residual binding of 1C4 is insufficient to effectively compete with ACE2. Taken together, these findings reveal the diverse neutralizing mechanisms deployed by antibodies against different variants and suggest that antibody escape results from a combination of decreased affinity and impaired ACE2-blocking capacity.

## Nanoparticle-based antibody polymerization rescues the efficacy of 1C4

Given that the diminished, but not abolished, binding efficacy of 1C4 against the XBB spike results in the loss of neutralization, we sought to engineer a polymerized form of 1C4 to recover its binding efficacy by increasing its binding avidity. Previous studies have demonstrated that the increasing antibody valency is a promising strategy for improving antibody avidity [22,31]. Here, we developed an easy-to-use strategy for polymerizing nAbs by fusing 1C4 with a SpyTag and conjugating it to the mi3 nanoparticle fused with a SpyCatcher (Fig 4A). The resultant 1C4-SpyTag was solubly expressed and covalently conjugated to nanoparticle mi3-SpyCatcher (mi3-1C4) with high efficiency, resulting in a molecular assembly of mi3-1C4 with a mass exceeding that of the mi3-SpyCatcher alone (Fig 4B–4F). The purity and homogeneity of the assembled mi3-1C4 nanoparticles were confirmed using high-performance size-exclusion chromatography (HPSEC), dynamic

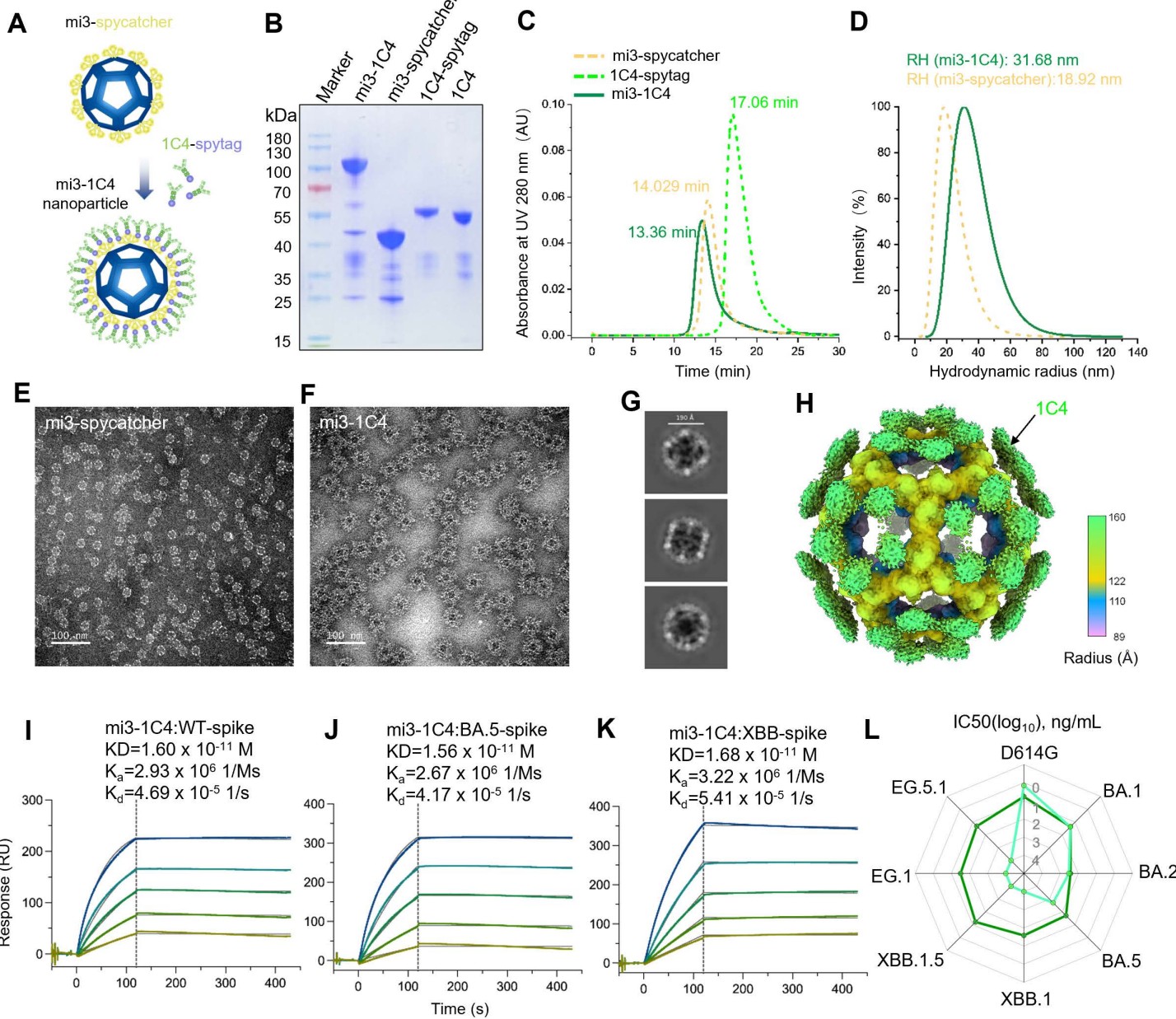

**Fig 4. The nanoparticle-based polymerization of 1C4 rescues the neutralization against the XBB variant. (A)** Schematic diagram of antibody 1C4 carrying SpyTag conjugated into the nanoparticle of mi3 decorated with SpyCatcher. These models were manually illustrated using paint tool SAI. **(B)** The identification of the polymerized 1C4 by reduced SDS-PAGE. **(C)** Characterization of the purified mi3-SpyCatcher, 1C4-SpyTag and mi3-1C4 by HPSEC analysis. **(D)** Detected diameter of mi3-SpyCatcher and mi3-1C4 showed enlarged dimeter of ~31.68nm for the latter. **(E and F)** The morphology and uniformity of mi3-SpyCatcher (E) and mi3-1C4 (F) were identified by negative staining TEM. Scale bar: 100nm. **(G and H)** Representative cryo-EM 2D classification results (G) and 3D reconstruction density map (H) of mi3-1C4. The outermost densities corresponding to the decorated 1C4 are indicated and colored in green. **(I-K)** The binding affinities of mi3-1C4 to the spike proteins of WT **(I)**, BA.5 (J) and XBB (K) were determined by surface plasmon resonance. Colored curves are the experimental traces obtained from surface plasmon resonance (SPR) experiments, and curves indicated the best local fit for the data are used to calculate the KD values by using a 1:1 binding model. **(L)** Radar plot based on the IC50 ($\log_{10}$) values demonstrating the neutralization profile of mi3-1C4 to SARS-CoV-2 D614G strain and diverse Omicron subvariants. 1C4-SpyTag served as control.

light scattering (DLS) assays and negative-staining transmission electron microscopy (NS-TEM) (Fig 4C–4F). DLS revealed an increased hydrodynamic radius for mi3-1C4 (31.68 nm) than the unlinked mi3-SpyCatcher (18.92 nm) (Fig 4D), directly corroborating the successful assembly of the polymerized nanoparticles. Furthermore, cryo-EM 2D classification (Fig 4G) and 3D reconstruction (Fig 4H) of mi3-1C4 confirmed the successfully decorating of 1C4 on mi3 nanoparticles; however, the densities of 1C4 on mi3-1C4 are smeared due to the flexible linking between 1C4-SpyTag and mi3-SpyCatcher.

To assess the enhanced binding affinity achieved with the nanoparticle mi3-1C4, we performed SPR assays to measure its affinity to the WT, BA.5, and XBB spikes. The kinetics data revealed that the nanoparticle mi3-1C4 exhibits a markedly enhanced affinity for all tested spike proteins compared with the unconjugated 1C4-SpyTag (Figs 4I-4K and S12). Notably, although 1C4 and 1C4-SpyTag showed low affinity for the XBB spike protein (8,750 nM and 1,480 nM, respectively) (S1 and S12 Figs), the polymerized mi3-1C4 improved both the association and dissociation rates, achieving a sub-nanomolar affinity with a KD value of 16.8 pM, comparable to its affinity observed for the WT (16 pM)) and BA.5 (15.6 pM) spikes (Fig 4I–4K). These findings suggested that the binding efficacy of the nanoparticle-based polymerized antibody is unaffected by SARS-CoV-2 variants.

We subsequently elucidated the neutralizing potency and breadth of the nanoparticle mi3-1C4 against the D614G strain and several Omicron subvariants, including BA.1, BA.2, BA.5, XBB.1, XBB.1.5, EG.1 and EG.5.1. As expected, compared with the monomeric 1C4-SpyTag that only neutralized BA.5 and preceding variants, the polymerized mi3-1C4 markedly enhanced its neutralizing potency and breadth (Figs 4L and S13). Specifically, while mi3-1C4 did not amplify the neutralizing efficiency against the D614G strain, it preserved strong neutralization with an IC50 value of 6 ng/mL. Additionally, mi3-1C4 showed neutralizing capabilities comparable to 1C4-SpyTag against BA.1 and BA.2, while it significantly bolstered neutralization against BA.5 (IC50: 52 ng/mL), exhibiting approximately a 10-fold inprovement of IC50 value (Fig 4L). Moreover, it reinstated effective neutralization against XBB.1, XBB.1.5, EG.1 and EG.5.1, with IC50 values ranging from 17-38 ng/mL (Figs 4L and S13). The enhanced and restored of neutralization capabilities of mi3-1C4 are in alignment with its high affinity to all tested spikes, underscoring the critical role of substantial binding affinity in effective neutralization. In summary, this results highlighted the success of the antibody modification in countering immune evasion. Taken together, our findings support the hypothesis that enhancing antibody multi-valency can not only improve the binding affinity and broaden the neutralizing breadth of antibody.

**mi3-1C4 provide effective protection against infections of Beta and XBB variants in hamster**

To evaluate the *in vivo* efficacy, mAb-1C4 and mi3-1C4 were tested in a hamster model challenged with either SARS-CoV-2 Beta or XBB variants. Specifically, hamsters were intranasally infected with $1 \times 10^4$ PFU of virus and treated at 1 day post-infection (dpi) with either 20 mg/kg mAb-1C4 or mi3-1C4 via intraperitoneal injection, while untreated animals served as controls. Survival and body weight were monitored daily until 7 dpi, at which point all surviving animals were euthanized for respiratory tissue collection (Fig 5A). After SARS-CoV-2 Beta variant infection, 2 out of 6 of the untreated hamsters survived at 7 dpi, whereas all of the hamsters with mAb-1C4 and mi3-1C4 survived (Fig 5B, left). The mi3-1C4 treated hamsters in group 3 showed a slightest of 5.55% body weight loss at 7 dpi, lower than that of 1C4-treated (11.12%) and untreated hamsters (23.28%) (Fig 5B, right), indicating that mi3-1C4 showed superior treatment than 1C4 against the infection of Beta variant in hamsters. Similarly, against the XBB variant, all of the hamsters in group 4–6 survived (Fig 5C, left), but mi3-1C4-treated hamster showed lower body weight loss of 2.8% at 7 dpi than 1C4-treated (6.53%) and untreated hamsters (11.86%) (Fig 5C, right). Of note, although 1C4 lost the *in vitro* neutralization against the infection of the XBB variant, it still showed protective potential against the infection of this virus in hamster to some extent, albeit with reduced potency compared to mi3-1C4. This residual *in vivo* protection may be attributed to a vestigial affinity for the XBB spike, whereas mi3-1C4 displayed superior efficacy.

The gross images of lung tissues collected from hamsters showed that both the treatments of mAb-1C4 and mi3-1C4 reduced the severity of lung injury caused by both SARS-CoV-2 Beta and XBB variants (Fig 5D and 5F). Viral RNA

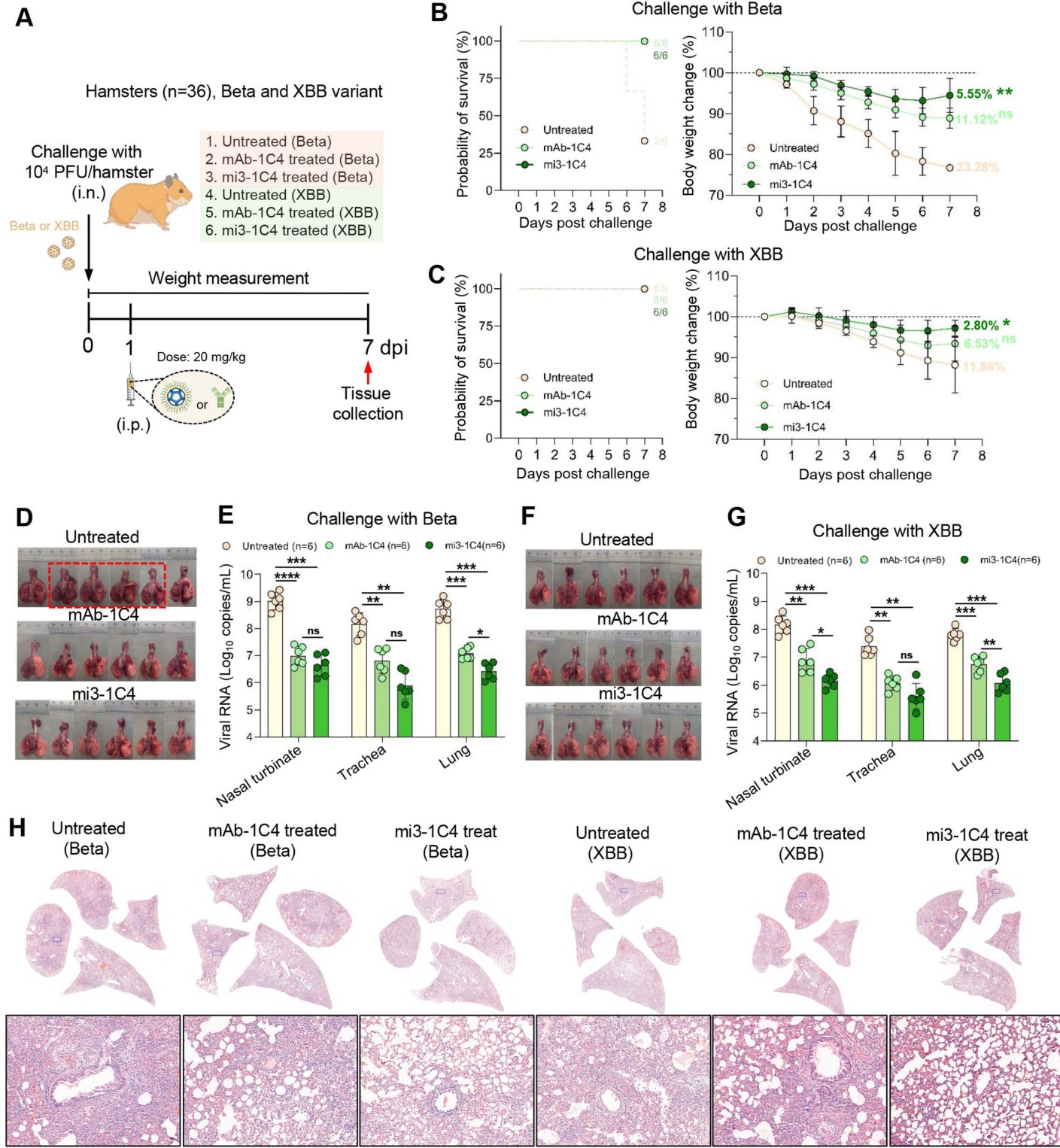

**Fig 5. mi3-1C4 provides enhanced and robust protection against the infection of both Beta and XBB variants in hamster. (A)** Scheme of virus infection, antibody therapy and sample collection in hamster model. Schematics were generated using BioRender (created in BioRender. Xia, F. (2025) https://BioRender.com/gj297eh) and manually illustrated using paint tool SAI. **(B and C)** Survival analysis (left panel) and record of body weight loss

(right panel) from 0 to 7 dpi (n = 6/group) for the treatment against the infection of the Beta (B) and XBB (C) variants. **(D)** Gross images of lung tissues collected at 7 dpi for the Beta variant infection groups. The lung tissues of hamsters die in the infection course are marked in red frame. **(E)** Viral RNA levels in nasal turbinate, trachea, and lung from the Beta variant infection groups were measured by RT-PCR, respectively (n = 6/group). Primers targeting the SARS-CoV-2 ORF1ab gene were used for amplification. Significance was determined by one-way ANOVA. Two-sided p-values <0.01 were considered significant: *P < 0.01, **P < 0.001, ***P < 0.0001, ns indicates no significance. **(F)** Gross images of lung tissues collected at 7 dpi for the XBB variant infection groups. **(G)** Viral RNA levels in nasal turbinate, trachea, and lung from the XBB variant infection n groups were measured by RT-PCR as that in (E). **(H)** Representative histopathological examination of hamster lung tissues.

levels in respiratory organs, including the nasal turbinate, trachea, and lungs, were quantified using RT-PCR targeting the SARS-CoV-2 ORF1ab region. Following Beta variant infection, the mi3-1C4-treated hamsters displayed the lowest viral RNA levels of $6.65 \pm 0.40$, $5.90 \pm 0.51$, and $6.42 \pm 0.31$ $\log_{10}$ (copies/mL) in nasal turbinate, nasal trachea and lung, respectively, significantly lower than that from untreated hamsters ($9.02 \pm 0.32$, $8.17 \pm 0.42$ and $8.71 \pm 0.35$ in nasal turbinate, nasal trachea and lung, respectively) (Fig 5E). Consistently, after SARS-CoV-2 XBB variant infection, the hamsters in mi3-1C4 treated group also showed lowest viral RNA levels of $6.17 \pm 0.24$, $5.61 \pm 0.45$, and $6.08 \pm 0.32$ $\log_{10}$ (copies/mL) in nasal turbinate, nasal trachea, and lung, respectively, compared to $6.89 \pm 0.42$, $6.08 \pm 0.26$, and $6.76 \pm 0.28$ $\log_{10}$ (copies/mL) for the 1C4 treated hamsters and $8.18 \pm 0.31$, $7.39 \pm 0.36$, and $7.81 \pm 0.21$ $\log_{10}$ (copies/mL) for the untreated hamster (Fig 5G). Overall, both mAb-1C4 and mi3-1C4 therapies effectively alleviated SARS-CoV-2 infection and pathology. Notably, polymerized mi3-1C4 exhibited superior efficacy in reducing body weight loss and viral RNA levels in respiratory tissues compared to mAb 1C4.

We next evaluated the protective efficacy of the two-verse antibody against lung damage at 7 dpi. Histopathological analysis of lung tissues at 7 dpi further supported the protective efficacy of the antibody treatments. Hematoxylin and eosin (H&E) staining revealed that lungs from untreated hamsters infected with either the Beta or XBB variant exhibited severe pathological changes, including diffuse alveolar damage, inflammatory cell infiltration around bronchi and vessels, and thickened alveolar septa (Fig 5H). In contrast, hamsters treated with mAb-1C4 displayed partially preserved alveolar structures and moderate inflammation. Notably, mi3-1C4 effectively prevented the development of lung pathology, with markedly improved morphology, intact alveolar spaces and only mild peribronchiolar and perivascular infiltration. These findings demonstrated that mi3-1C4 provides enhanced protection against virus-induced pulmonary injury, consistent with its superior antiviral efficacy observed in viral load reduction and body weight maintenance.

### mi3-1C4 promotes spikes aggregation and viral cross-linking

To better understand mi3-1C4's marked antiviral activities, we performed negative-staining transmission electron microscopy (NS-TEM) to visualize the complex formation between spike proteins and antibodies. Specifically, the WT, BA.2, and JN.1 spike trimers were individually incubated with either conventional IgG-format 1C4 or multimerized mi3-1C4. The NS-TEM images showed that incubation of the WT spike trimers with 1C4 led to visible spike aggregation, whereas the BA.2 and JN.1 spikes remained dispersed in the presence of 1C4 (Fig 6A–6B). This finding supports the notion that even though 1C4 may not fully lock all WT spike particles into the closed conformation, it can compensate for this limitation by inducing spike aggregation and thereby blocking ACE2 binding. In contrast, mi3-1C4 induced robust aggregation of spike trimers of both BA.2 and JN.1 variants (Fig 6C–6D), demonstrating broader cross-linking capabilities and higher functional avidity. This observations suggest that mi3-1C4 may endow mechanism by which the polymerized 1C4 can induce inter-spike cross-linking that impair virus-host interactions by preventing receptor engagement (Fig 6E). We further performed cryo-EM imaging of native SARS-CoV-2 virus (Victoria) incubated with either mi3-1C4 or 1C4 to assess the viral cross-linking capacity of mi3-1C4 (Fig 6F–6I). Our results indicate that while 1C4 sometimes links two viral particles (Fig 6H), mi-1C3 induces inter-spike cross-linking of multiple SARC-CoV-2 viruses and form large aggregates (Fig 6I), which may impair virus-host interactions (Fig 6J). This structural model illustrates how multivalent antibody architecture

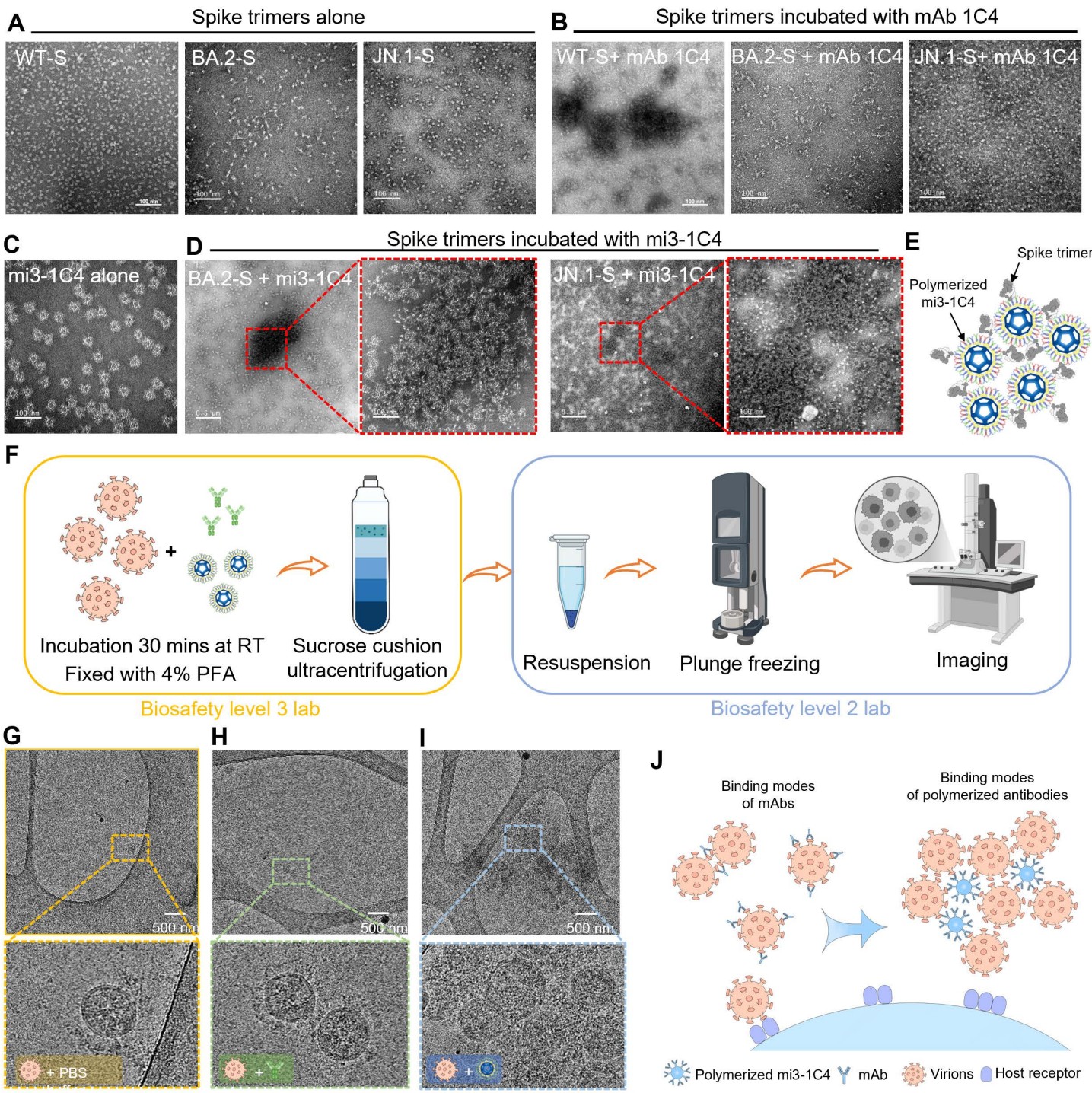

**Fig 6. mi3-1C4 promotes aggregation of SARS-CoV-2 spikes and cross-linking of native virions. (A and B)** Representative negative-staining electron microscopy images of WT, BA.2, and JN.1 spike proteins alone (A), in complex with 1C4 (B). **(C and D)** Negative-staining images of mi3-1C4 alone (C) and after incubation with spike trimers at room temperature for 30 minutes (D), demonstrating spike aggregation triggered by multivalent binding. **(E)** Schematic representation of the interaction between the spike trimers and polymerized mi3-1C4. **(F)** Experimental pipeline for evaluating the effects of mi3-1C4 treatment on native SARS-CoV-2 virions. The samples were fixed prior to transfer from the BSL-3 facility to BSL-2. **(G-I)** Cryo-EM analysis indicates mi3-1C4 promotes the formation of aggregated virions. Representative cryo-EM raw images of native SARS-CoV-2 virions alone (G), virion:1C4

complexes (H) and virion:mi3-1C4 complexes (I) are shown. **(J)** Cartoon illustration depicting the proposed neutralizing mechanism of polymerized mi3-1C4 through cross-linking of virions. Schematics were generated using BioRender (created in Created in BioRender. Xia, F. (2025) https://BioRender.com/gj297eh) and manually illustrated using paint tool SAI.

transforms binding events into higher-order aggregation that sterically blocks viral entry. Crucially, this enhanced cross-linking capacity represents a distinct advantage over conventional IgG antibodies, which rely primarily on Fab-RBD binding but lack the spatial architecture to induce extensive virion clustering.

## Discussion

Since the emergence of the SARS-CoV-2 BA.2.86 lineage, particularly the JN.1 sublineages, as well as related variants such as NB.1.8.1 (a derivative of XDV.1.5.1), KP.3, and LP.8.1.1, the virus has undergone rapid evolution and widespread transmission, surpassing the previously dominant XBB subvariants [8,32–34]. The ancestral Omicron and subsequent XBB subvariants exhibit marked resistance to neutralization by sera from prior infection or vaccination due to extensive spike protein mutations [7,35,36]. Consequently, breakthrough and recurrent infections after vaccination have intensified the health burden, particularly for the elderly and immunocompromised individuals prone to complications. This underscores the urgent need for developing broad-spectrum neutralizing antibodies and elucidating the mechanisms of antibody escape and epitope evolution.

The emergence of SARS-CoV-2 variants with neutralizing antibody escape underscores the remarkable adaptability of the virus and highlights the urgent need to reconsider current vaccine and therapeutic strategies. Novel variants with mutations, particularly in the RBD of spike protein, poses a pressing challenge for the neutralizing capacity of pre-existing antibodies. These antibodies, whether elicited by vaccination, previous infection, or administered as part of therapeutics, were primarily designed to target epitopes based on the original strain or early variants of SARS-CoV-2. Consequently, the affinity and effectiveness of these antibodies, as exemplified by 1C4 in this study, can be compromised when confronted with newly variants. Furthermore, it has been suggested that the evolutionary progression of SARS-CoV-2 is shaped by synergistic forces, particularly the dual pressures of immune escape and ACE2 receptor affinity [37]. Most Omicron subvariants, such as BA.2.75, exhibit a markedly increased binding affinity for the ACE2 receptor compared to the ancestral SARS-CoV-2, which is likely associated with their enhanced transmissibility [38]. In contrast, certain subvariants, including XBB and XBB.1, show a modest reduction in ACE2 binding affinity compared to earlier lineages like BA.2 and BA.5 [10]. Nonetheless, the newly emerged XBB.1.5 and BA.2.86 variants maintain an increased affinity for the ACE2 receptor [5,39]. Despite these differences in receptor binding, a consistent pattern of increased immune evasion is observed across these variants. This trend is likely driven by mutations that diminish antibody binding without substantially weakening virus–receptor interactions. As such, accurate molecular and comprehensive analysis of antibody–receptor dynamics are essential for elucidating mechanisms of immune escape and guiding the development of next-generation therapeutics.

The neutralization mechanisms of most SARS-CoV-2 nAbs primarily relies on the high-affinity binding to the spike protein, ensuring occupation of the RBD and barring ACE2 access. Given that most Omicron mutations on the RBD are located at or near the receptor binding motif (RBM) but maintain sufficient or higher affinity to ACE2 receptor, it is not surprising Omicron variants would easily compromise the efficacy of those RBM-specific nAbs (e.g., 8H12 in this study). Namely, this observation offers indirect support for the previously proposed notion that immune escape and ACE2 affinity collaboratively drove the evolution of SARS-CoV-2 [37]. In this study, we identified a canonical Class 3 nAb, 1C4, engaging a relative conservative region of the RBD of SARS-CoV-2 non-overlapping to the RBM. 1C4 effectively neutralizes SARS-CoV-2 D614G strain and multiple earlier variants but is escaped by BQ.1.1 and later XBB sublineages. 1C4 targets a conserved non-RBM epitope on the RBD. However, unlike typical Class 3 nAbs, 1C4 can sterically hinder ACE2 binding by engaging all closed RBDs on the spike trimer simultaneously, offering an alternative mechanism of neutralization.

Intriguingly, such RBD "locking" capacity of 1C4 is no longer exist when binding to the BA.5 spike, with evidence of the immune complex structure of BA.5-S:1C4 showing the accessible to ACE2 binding due to remaining at least one RBD in open state. Unlike antibodies such as BG10–19, which bridge two adjacent RBDs and sterically restrict the transition of RBDs from "down" to "up" conformation [40], 1C4 and other similar antibodies like 002-S21F2 [41], display a different binding mode where each Fab engages a single RBD independently. Previous evidence have suggested that the Omicron spikes exhibit reduced structural heterogeneity, often adopting a single dominant conformation [16]. This aligns with our cryo-EM structures of WT-S:1C4 which revealed two binding patterns: one involving both up and down RBDs, and another with all RBDs in the down state. These structural studies indicated that the Omicron mutations may enhance RBD conformational dynamics and reduce the effectiveness of antibodies that rely on conformational locking. On the other hand, we observed that bivariant 1C4-IgG could promote spike cross-linking upon binding to the WT spikes. This explain why 1C4 still retains ACE2-blocking activity even though not all WT-S:1C4 particles adopt the "lockdown" conformation. A similar observation was found for antibody 6H2-IgG, which binds to the spikes to form a head-to-head trimers dimerization linking, resulting in ACE2 inaccessibility and neutralization outcomes [42]. In contrast, for the BA.2 and later Omicron variants, 1C4 fails to lock the spike in the closed conformation and even does not induce spike-antibody aggregation. Therefore, the neutralization of Omicron subvariants by 1C4 likely involves alternative mechanisms, such as inhibition of membrane fusion or prevention of S1 shedding, which have also been proposed for antibodies like SP1–77 [43]. These findings together elucidate the complexity of antibody-mediated neutralization and the diverse strategies employed by viruses to evade immune responses.

To clarify the molecular basis of how antibody 1C4 evades neutralization, we conducted a comprehensive investigation and employed cryo-EM to visualize an ensemble of diverse spikes of SARS-CoV-2 variants in complex of 1C4 and a 1C4-based triple-antibody (1C4/8H12/3E2). The inclusion of 3E2 and 8H12 alongside 1C4 allowed us to examine how distinct antibody epitopes respond differently to viral evolution. Structural analyses revealed a common feature: all three antibodies show markedly reduced binding against the certain Omicron subvariants, yet residual interactions remain detectable, indicating that viral escape does not necessarily eliminate binding. The key difference lies in their epitope locations, which determine each antibody's degree of sensitivity and resilience to mutations. Notably, 1C4 retains measurable binding to all tested spikes, including XBB, but its affinity is substantially weakened, preventing stable engagement of the RBD and abolishing neutralization. These findings highlight that loss of neutralization does not always equate to loss of binding. Given the retained affinity of the newly emerging variants such as XBB, we propose that antibody engineering of 1C4 could restore or enhance its binding affinity, potentially reviving its neutralizing capability. We and other studies have demonstrated that such nanoparticle displaying of both antibodies and antigens can enhance efficacy effectively [22,44]. Such an approach could significantly improve antibody responses and inform vaccine design for rapidly mutating viruses such as SARS-CoV-2 and HIV [31,45]. In this study, we have developed and validated a nanoparticle-based 1C4 (mi3-1C4), which demonstrates high affinity towards all evaluated spikes, including XBB. Notably, this enhanced binding occurs even as the affinity of the parental nAb 1C4 for the XBB RBD is significantly diminished. However, what exactly is the reason that antibody polymerization based on nanoparticles possesses such ultra-high affinity and broadly neutralizing capability? To unravel the hidden mechanisms, we prepared the sample of mi3-1C4 in complex of both the trimeric spikes (BA.2-S and JN.1-S) and native SARS-CoV-2 virus, for observation via electron microscopy. As expected, this result revealed that mi3-1C4 capitalizes on the multivalency of antibodies—potentially offering up to 120 binding sites—thereby markedly enhancing antigen contact probability and explaining the observed high affinity. Crucially, mi3-1C4 prompted an aggregation effect when binding to the S trimer, and induced viral cross-linking when binding to the native virus. These phenomenon, akin to the cross-linked S trimer induced by the IgG version of the certain nAbs [17,41], and the viral cross-linking potency of a recent reported modular nanobody AMETA [25], might introduce steric hindrance that impedes ACE2 interaction, thus facilitating robust and broad neutralization. Therefore, these advantages enable multimeric antibodies to withstand the effects of mutation sites, allowing existing antibodies with no or low neutralizing capacity to

enhance or restore potent neutralization capabilities. This will significantly reduce the time required for repeated antibody screening, providing a new approach for the development of broadly neutralizing antibodies.

A key limitation of this study is that we did not evaluate the potential immunogenicity of the mi3 scaffold, either on its own or as part of the mi3-1C4 complex. Pascha et al. [46] reported that neuramindase (NA)-mi3 elicited higher titers of anti-mi3 antibodies compared to mi3 alone. There is currently no evidence suggesting that antibody multimerization enhances the immunogenicity of the mi3 scaffold. Despite these concerns, the mi3 platform offers a unique advantage in its high valency. Compared to ferritin (which displays 24 antigens) [31] and AMETA (over 20 nanobodies) [25], the mi3 scaffold can theoretically present up to 60 antibody fragments. This increased multivalency substantially enhances binding avidity and cross-linking potential, making it especially valuable for rescuing antibodies with weak or partially compromised binding to antigenic variants, as demonstrated in our study. Furthermore, advances in computational protein design have accelerated the development of nanoparticle-based antibody platforms. Rational engineering or surface humanization of the mi3 scaffold may mitigate immunogenicity and improve translational potential. Given its modularity, mi3 may also be broadly applied to other viral targets beyond SARS-CoV-2, supporting generalizable strategies for antibody rescue and enhancement.

In summary, this study conducted a comprehensive evaluation of the changes in binding affinity and neutralization capacity of antibodies against various SARS-CoV-2 variants utilizing biochemical and structural approaches. We also explored ACE2 inhibition and conducted structural investigations, uncovering varied neutralization strategies of 1C4 tailored to each variant. Our findings confirmed, at the molecular level, the alterations in antibody interactions with Omicron subvariants, informing strategies to augment antibody affinity. This work underscores the importance of binding affinity in targeting Omicron variants and supports the effectiveness of optimizing antibody affinity. Furthermore, our findings lay a theoretical groundwork for developing antibodies to tackle emerging variants.

## Materials and methods

### Ethics statement

The Syrian Hamster was bred in a specific pathogen-free (SPF) facility. All animal experiments were approved by the Medical Ethics Committee (XMULAC20210131). Studies involving infectious SARS-CoV-2 were conducted in biosafety level 3 (BSL-3) and animal biosafety level 3 (ABSL-3) laboratories. Personnel adhered to strict biosafety protocols, including the use of powered air-purifying respirators and disposable coveralls when handling the virus and animals in isolators. Decontamination procedures were rigorously followed, including disinfection and mandatory showering upon exiting. The facility, procedures, training records, safety drills, and inventory logs underwent regular inspection and oversight by institutional biosafety officers in consultation with facility managers.

### Cell lines

The cell lines utilized in this study were procured from ATCC (H1299, 293T, and Vero) and Thermo Fisher Scientific (Expi293F). All cell lines were regularly tested for mycoplasma contamination and confirmed to be mycoplasma-free.

### The expression and purification of spike proteins

The spike proteins (residues 1–1208) and the RBDs (residues 319–541) of SARS-CoV-2 variants were obtained as previously reported [47,48]. In brief, a gene encoding the ectodomain of a prefusion conformation-stabilized S protein (Gen-Bank: MN908947, GenBank: MN908947 for SARS-CoV and SARS-CoV-2 S genes, respectively) with proline substitutions at 986 and 987, 'GSAS' substitution at the S1/S2 furin cleavage site (residues 682–685), a C-terminal T4 fibritin trimerization motif, an HRV3C protease, and 8×His-Tag was synthesized and individually cloned into the pcDNA3.4 vector. The genes of other trimeric spike (WT-S6P, BA.1-S6P, BA.2-S6P, BA.2.75-S6P, BA.5-S6P, XBB-S6P) and RBD (WT-RBD,

BA.2.75-RBD, BA.5-RBD) proteins were also synthesized, of which the S6P version contained an additional four Pro substitutions (F817P, A892P, A899P, and A942P) compared with S2P. The other recombinant spike proteins (Alpha-RBD, Beta-RBD, Delta-RBD, BA.1-RBD and BA.2-RBD) were purchased from Sino Biological Inc.

Recombinant protein expression was performed using Expi293F cells. Briefly, plasmids encoding for targeted proteins were transiently transfected into Expi293 cells using PEI MW40,000 (Yeasen). Cell-free supernatants were obtained 7 days after transfection by centrifugation and filtration with a 0.22-μm filter. Subsequently, the proteins were purified using a Ni-Sepharose 6 Fast Flow (Cytiva) column and stored in PBS buffer.

### The expression and purification of mi3-SpyCatcher

The mi3-SpyCatcher was obtained as previously reported [22]. Briefly, mi3-SpyCatcher particles were produced in E.coli BL21 (DE3) RIPL cells (Agilent) transformed with a pET28a-mi3-SpyCatcher plasmid featuring an N-terminal 6×His tag and purified using the following steps. After harvesting the bacteria via centrifugation, they were resuspended in 10 mL of lysis buffer (25 mM Tris–HCl pH 8.5, 300 mM NaCl, 0.1 mg/mL lysozyme, 1 mg/mL cOmplete mini EDTA-free protease inhibitor (Merck), and 1 mM phenylmethanesulfonyl fluoride (Sigma-Aldrich)), lysed by sonication in ice, and further centrifuged at 16,900×g for 30 min at 4°C. The supernatant was filtered (0.45 μm and 0.22 μm filters), precipitated with saturated ammonium sulfate at 4°C for 1h with stirring at 100 rpm, and the resulting particles were pelleted at 30,000×g for 30 min at 4°C. The pellet was resuspended in TBS pH 8.5 (25 mM Tris–HCl, 150 mM NaCl), dialyzed overnight at 4°C against TBS pH8.5, and finally purified by Superose 6 (Cytiva).

### Assembly and purification of mi3-1C4

The mi3-1C4 nanoparticles were assembled through the mi3-SpyCatcher particles incubated with 1C4-SpyTag at 1:3 molar ratio overnight at 4°C in TBS pH8.5. The conjugated mi3-1C4 particles were purified by size-exclusive chromatography (SEC) on a Superose 6 (Cytiva) equilibrated with TBS pH 8.5 (25 mM Tris–HCl with 150mM NaCl) and analyzed by SDS-PAGE.

### Expression of antibodies and fabs

8H12, 3E2, and 1C4 were obtained by mouse hybridomas preparation as described previously [49]. Ascites of the three nAbs were prepared by injecting hybridoma cells into the peritoneal cavities of pristine-primed BALB/c mice, and were collected 9–12 days later and stored at −20 °C. To prepare 1C4-SpyTag, the heavy and light chains of 1C4-SpyTag were cloned into pTT5-H and pTT5-K vectors (Youbio), respectively. The 1C4-SpyTag heavy chain contains the heavy chain sequence, a (GGGGS)2 spacer and SpyTag sequence. The paired heavy and light chain expression cassettes were transiently co-transfected into Expi293F cells with equal amounts of plasmids using PEI MW40,000. All murine (ascites) and recombinant 1C4-SpyTag were purified by affinity chromatography using MabSelect Sure resin (Cytiva) and were stored in PBS. To further prepare the Fab fragments, mAbs 8H12, 3E2, and 1C4 were digested with papain at 0.1% (w/w) in PBS containing 30 mM l-cysteine and 50 mM EDTA at 37 °C for 12 h. Digestion was terminated with the addition of 20–30 mM iodoacetamide.

### Pseudovirus neutralization assay based on lentiviral pseudovirus

The neutralizing capacities of nAbs against the SARS-CoV-2 ancestral strain and related variant strains were tested using lentiviral (LV) pseudotyping particles bearing the S protein, as previously described [50]. Briefly, lentiviral pseudovirions carrying the S protein were produced by co-transfecting a lentiviral packaging plasmid (psPAX2, Addgene), S plasmids of SARS-CoV-2 variants, and a green fluorescent protein (mNeonGreen) reporter vector (pLvEF1α-mNG, containing an EF1α promoter-driven mNeonGreen expression cassette) in 293T cells. Mixtures of serially diluted nAbs and LV pseudotyping particle inoculum (0.5 TU/cell) were incubated for 1 hour before being transferred into 96-well cell culture plates

with optically clear bottoms, pre-seeded with H1299-ACE2hR cells (H1299 cells stably overexpressing human ACE2 and nuclear-localized H2B-mRuby3) for 36-hour incubation. Fluorescence images were subsequently collected using Opera Phenix or Operetta CLS high-content equipment (PerkinElmer) and quantitatively analyzed with Columbus Software 2.5.0 (PerkinElmer). To determine antibody neutralizing activity, the reduction percentage of mNeonGreen-positive cells in nAb-treated wells compared to control wells was calculated. IC50 values were determined using the 4-parameter logistic (4PL) regression in GraphPad Prism (version 8.0.1) (https://www.graphpad.com).

### Affinity determination by surface plasmon resonance (SPR)

The binding affinities of the mAbs to different spike-trimer proteins or RBDs were determined by SPR assays using a Biacore 8K instrument (GE Healthcare). The viral antigen proteins (S6P or RBD) were covalently amine-coupled to CM5 sensor chips, respectively. Serially diluted 1C4 (800, 400, 200, 100, 50, 25, 12.5 and 6.25 nM) and mi3-1C4 (6.7, 3.35, 1.675, 0.8375, 0.41875 and 0.209375 nM) were then flowed through the sensor surface at a flow rate of 30 μL/min in PBS-P+ buffer (0.2 M phosphate buffer with 27 mM KCl, 1.37 M NaCl, and 0.5% Tween-20). The flow durations were 120 s for the association stage and 300 s for the dissociation stage. Finally, the affinity was calculated using a 1:1 binding fit model with BIAevaluation software.

For competitive SPR, mAbs or hACE2 were diluted to 1600 nM in PBS-P buffer. Antibodies (first protein) were loaded onto the biosensors for 180 s binding, and then subjected to flow of a second interacting protein (hACE2 or the second antibody) for another 500 s, or even a third antibody to determine simultaneous binding of three mAbs (e.g., 8H12+3E2+1C4). The unblocked pattern of the spike protein in buffer was used as a control.

### Negative-staining electron microscopy

The purified mi3-SpyCater, mi3-1C4, WT-, BA.2- and JN.1-spike and their complexes were diluted in PBS and then absorbed onto 200 mesh carbon-coated copper grids (Quantifoil Micro Tools) for 30 s. Grids were washed twice with double-distilled water and negatively stained with 2% phosphotungstic acid (pH 6.4) for 30 s. Specimens were evaluated and imaged with a FEI Tecnai T12 transmission electron microscope (TEM).

### Cryo-EM sample preparation and data collection

Aliquots (3 μL) of 3 mg/mL mixtures of purified SARS-CoV-2 spike trimers of Omicron subvariants in complex with excess Fab fragments of 1C4 or 1C4/8H12/3E2 were incubated in 0.01% (v/v) Digitonin (Sigma) and then loaded onto glow-discharged (60 s at 20 mA) holey carbon Quantifoil grids (R1.2/1.3, 200 mesh) using a Vitrobot Mark IV (Thermo Fisher Scientific) at 100% humidity and 4°C. Data were acquired with the SerialEM software on an FEI Tecnai F30 transmission electron microscope (Thermo Fisher Scientific) operated at 300 kV and equipped with a Gatan K3 direct detector. Images were recorded in the 36-frame movie mode at a nominal 39,000 × magnification in super-resolution mode with a pixel size of 0.389 Å. The total electron dose was set to 60 e$^-$/Å$^2$, and the exposure time was 4.5 s.

As for WT-RBD complexed with 1C4 and ACE2, 1C4 and ACE2 were incubated with an excess amount of RBD at room temperature for 0.5 h followed by purification via Superdex 200 (Cytiva). The purified complex was then prepared as Cryo-EM sample preparation using the same method described above. The image datasets were collected on the Titan Krios G4 equipped with a Gatan K3 and BioContinuum HD Imaging Filter operated in zero-loss mode with a slit width of 15 eV, at a nominal magnification of 130,000 x, which corresponds to a physical pixel size of 0.65 Å (0.325 Å in super-resolution mode).

### Image processing and 3D reconstruction

Drift and beam-induced motion correction were performed using MotionCor2 [51] to produce a micrograph from each movie. Contrast transfer function (CTF) fitting and phase-shift estimation were conducted with Gctf [52]. Micrographs

with astigmatism, significant drift, or contamination were discarded before reconstruction. The subsequent reconstruction procedures were performed using Cryosparc V3 [53]. In brief, particles were automatically picked using the "Blob picker" or "Template picker". Several rounds of reference-free 2D classifications were performed, and selected particles were then subjected to ab initio reconstruction, heterogeneous refinement, and final non-uniform refinement. The resolution of all density maps was determined by the gold-standard Fourier shell correlation curve, with a cutoff of 0.143 [54]. Local map resolution was estimated with ResMap [55].

### Atomic model building, refinement, and 3D visualization

The structure of the RBD from the WT trimeric spike (pdb no. 7XNS [16]) was employed as the initial model for the BA.5 RBD. Templates were initially fitted into the corresponding final cryo-EM maps using Chimera [56], followed by manual correction and adjustment through real-space refinement in Coot [57]. The resulting models were refined with phenix.real_space_refine in PHENIX [58]. This process was iteratively performed until problematic regions, Ramachandran outliers, and poor rotamers were either eliminated or moved to favored regions. Final atomic models were validated using Molpro-bity [49,50]. All Figs were generated with Chimera or ChimeraX [59,60].

### Site-directed mutagenesis and structural modeling

The BA.5-RBD structure was loaded into Discovery Studio 2017 R2 (DS) software, and targeted amino acid substitutions were performed using the Build Mutant tool. The target residue for mutation was selected and replaced with the desired amino acid variant from the rotamer library. The optimal rotamer conformation was selected based on steric clash minimization and hydrogen bonding potential. Subsequently, energy minimization was performed on the mutated structure using the CHARMM36 force field integrated in DS, with a maximum of 2,000 steps or until a convergence threshold of 0.1 Å RMSD was reached. The final structural models were validated through Ramachandran plot analysis to ensure structural rationality.

### Blockage of spikes of SARS-CoV-2 and Omicron subvariants binding to hACE2

Antibodies were three-fold serially diluted from an initial concentration of 10 µg/mL and mixed with equivoluminal hACE2 tagged with human IgG Fc (final concentration 1.5 µg/mL) for 1 h at 37°C. These mixtures were then added to the wells of 96-well plates pre-coated with diverse S6P proteins, and incubated at 37 °C for 1 h. ACE2 mixed with ELISA buffer without antibody served as a control. Subsequently, the plates were washed and incubated with HRP-conjugated anti-human IgG to detect hACE2-Fc binding. The readout was detected at a wavelength of 450 nm. The final percentage inhibition values (%) were calculated as follows: $(100 - (OD_{sample}/OD_{control}) \times 100)$.

### Size-exclusive chromatography

1C4-SpyTag, mi3-SpyCatcher, mi3-1C4 were subjected to HPLC (Waters) analysis using a TSK Gel G5000PWXL 7.8 × 300 mm column (TOSOH) equilibrated in TBS pH 8.5. The system flow rate was maintained at 0.5 mL/min and eluted proteins were detected at 280 nm.

### Virus inoculation and sample collection

The SARS-CoV-2 Delta variant (EPI_ISL_2385091) and XBB.1.9.2.1 variant (EPI_ISL_17660518) were passaged on Vero cells (#CCL-81, ATCC), respectively. Viral stocks were prepared in Vero cells with DMEM containing 2% FBS, 5ug/mL TPCK-trypsin, 1% Penicillin Streptomycin and 30mmol/L $MgCl_2$. Viruses were harvested and stored in ultra-low temperature freezer. The titers were determined by means of plaque assay in Vero cells. As previous described [61–63], 10~20-week-old male hamsters were anesthetized with isoflurane (#R510-22, RWD Life Science) and intranasally

inoculated with $1 \times 10^4$ PFU of SARS-CoV-2 Beta or XBB variant in 200 μL PBS (#10010031, GIBCO). At 1 day post infection (dpi), animals received intraperitoneal injection of either 20 mg/kg mAb-1C4 (group 2, n = 6) or mi3-1C4 (group 3, n = 6). Survival and body weight were monitored daily for 7 days. At endpoint, respiratory tissues (turbinate, trachea, lung) were collected from surviving and deceased animals. Gross morphology of lung tissues was documented for preliminary assessment of lung injury.

### Detection of Viral RNA

For the solid organ samples, we collect ~1 g turbinate, ~ 0.1 g trachea and ~0.1 g lung in 1mL PBS for homogenate and detection of viral RNA and viral titer. Viral RNA was extracted by using a QIAamp Viral RNA Mini kit (#52906, Qiagen) according to the manufacturer's instructions. The RT-qPCR was conducted by using the SLAN-96S Real-Time System (Hongshi, Shanghai, China) with a SARS-CoV-2 RT-qPCR Kit from Wantai (Beijing, China). Relative Viral RNA of SARS-CoV-2 ORF1ab gene was determined using primer pairs and probes provided in the kit. Viral RNA copies were expressed on a log10 scale after normalized to the standard curve obtained by using ten-fold dilutions of a SARS-CoV-2 stock.

### SARS-CoV-2 Victoria variant culture and titration

SARS-Co-2 Victoria variant propagation, titration and binding to antibodies were performed in the containment level 3 (CL3) lab at the Oxford Particle Imaging Centre (OPIC). Victoria variant was a kind gift from Tao Dong's group, CAMS-Oxford Institute. Vero E6 cells was a kind gift from Ervin Fodor's group, Sir William Dunn School of Pathology, Oxford.

 15 mL of $6 \times 10^6$ Vero E6 cells were seeded into a T75 flask one day prior to infection. On the day of infection, the medium was replaced with 15 mL of DMEM supplemented with 1% FBS and Glutamine. 100 mL of virus at a titer of $10^5$ was added to the flask, and the culture was incubated for 2–3 days. Cell debris was removed by centrifugation at $400 \times g$ for 20 minutes at 4°C. The supernatant was aliquoted into 1 mL portions and stored at −80°C. After thawing, the virus titer was determined by plaque assay.

### Antibodies induction and Cryo-EM sample preparation

mi3-1C4 antibodies were prepared in 20 μL and added to 1 mL of supernatant of Victoria variant culture (titer = $5 \times 10^5$/mL) to achieve final concentrations of 1 μg/mL. PBS solution and 1C4 antibody (0.6 μg/mL in culture supernatant) were used as control groups. The mixtures were incubated at 37°C for 1 hour, then fixed with 4% paraformaldehyde (EM grade) at room temperature for 30 mins before being removed from the CL3 facility. Subsequently, the mixture was loaded onto a 1 mL 10% sucrose cushion in STE buffer and centrifuged at 30,000 rpm for 3 hours. The supernatant was carefully discarded, and 20 μL of PBS buffer was added to the centrifuge tube to gently re-suspend the pellet overnight.

 EM grids (R2/2 Quantifoil holey carbon, 300 mesh on copper) were glow-discharged prior to use. A 3.5 μL aliquot of re-suspended samples was applied to the carbon side of the grids. The grids were blotted from the back for 3 seconds and plunge-frozen in liquid ethane at −183°C using the Leica GP2 cryo-plunger. The grids were then clipped and screened at various magnifications using a 200 kV Thermo Scientific Glacios transmission electron microscope.

## Supporting information

**S1 Fig. The binding affinities of 1C4 against diverse SARS-CoV-2 variants.** (A) The binding affinities of 1C4 to the RBDs of spike proteins of SARS-CoV-2 WT, Alpha, Beta, Delta, BA.1, BA.2, BA.2.75, BA.5 and XBB variants were determined by surface plasmon resonance. Colored curves are the experimental traces obtained from surface plasmon resonance (SPR) experiments, and curves indicated the best local fit for the data are used to calculate the KD values by using

a 1:1 binding model or steady state affinity. (B) A summary of the KD values of 1C4 calculated from the SPR assays in (A). NA indicates not applicable.
(PDF)

**S2 Fig. The neutralization breadth of 1C4 against SARS-CoV-2 and its variants.** The IC50 values represent the antibody concentration required to achieve 50% neutralization of viral infection. Data were collected from three technical replicates and displayed as means±SD.
(PDF)

**S3 Fig. Single-particle cryo-EM data processing workflow for the immune complex of BA.5-S:1C4.** (A) Representative electron micrograph (scale bar: 50 nm), 2D classification results, heterogeneous refinement maps, and final refinement maps (colored by local resolution) are shown. (B-D) FSC curves for the reconstruction of state 1 by global refinement (B) and localized refinement (C), and state 2 by global refinement (D) are shown.
(PDF)

**S4 Fig. The intraction details of 1C4-paratope.** (A) Surface representation of the 1C4 Fab, with interacting residues highlighted as transparent sticks. The corresponding residues on the RBD that form the epitope are shown as purple sticks. Surface hydrophobicity map of the 1C4 paratope.(B) The detail of cation-π interaction between R346 and W102 of 1C4 heavy chain.
(PDF)

**S5 Fig. The influence of glycan on the binding of 1C4.** (A) SDS-PAGE analysis illustrates the spike protein RBDs with glycans and after treatment with PNGase F to remove glycans. (B) Impact of glycan removal on 1C4 binding to the RBD. The corresponding 50% effective concentration (EC50) is labeled.
(PDF)

**S6 Fig. Sequence alignment of SARS-CoV-2 variants.** The residues involved in 1C4-footprint highlighted in green.
(PDF)

**S7 Fig. Single-particle cryo-EM data processing workflow for the immune complexes of WT-S:8H12:3E2:1C4, BA.1-S:8H12:3E2:1C4 and BA.2-S:8H12:3E2:1C4.** (A-C) Representative electron micrograph (scale bar: 50 nm), 2D classification results, heterogeneous refinement maps, and final refinement maps (colored by local resolution) of WT-S:8H12:3E2:1C4 (A), BA.1-S:8H12:3E2:1C4 (B) and BA.2-S:8H12:3E2:1C4 (C) respectively, are shown. The representative FSC curves for the global reconstruction and localized refinement are also shown.
(PDF)

**S8 Fig. Single-particle cryo-EM data processing workflow for the immune complexes BA.** 2.75-S:8H12:3E2:1C4, BA.5:8H12:3E2:1C4 and XBB:8H12:3E2:1C4. (A-C) Representative electron micrograph (scale bar: 50 nm), 2D classification results, heterogeneous refinement maps, and final refinement maps (colored by local resolution) for BA.2.75-S:8H12:3E2:1C4 (A), BA.5:8H12:3E2:1C4 (B) and XBB:8H12:3E2:1C4 (C), respectively, are shown. the representative FSC curves for the global reconstruction and localized refinement are also shown.
(PDF)

**S9 Fig. Footprints comparison of 8H12, 3E2 and 1C4 on the BA.1 and BA.2 RBDs.** (A and B) The footprints of 8H12, 3E2 and 1C4 are colored by red, blue and green, respectively, on the surface of the BA.1 RBD (A) and BA.2 RBD (B). The BA.1 and BA.2 mutations are highlighted and the residues involved in epitopes of antibodies are labeled.
(PDF)

**S10 Fig. SPR-based sensorgram illustrates the non-competitive binding of 1C4 and ACE2 to the WT-RBD.**
(PDF)

**S11 Fig. Single-particle cryo-EM data processing workflow for the immune complexes of WT-RBD:1C4:ACE2 and WT-S:1C4.** Related to Fig 3. (A and B) Representative electron micrograph (scale bar: 50 nm), 2D classification results, heterogeneous refinement maps, and final refinement maps (colored by local resolution) are shown. (C and D) FSC curves for the reconstruction are shown.
(PDF)

**S12 Fig. The binding affinities of 1C4-SpyTag to the spike proteins of WT, BA.5 and XBB were determined by SPR.** Related to Fig 4. Colored curves are the experimental traces obtained from SPR experiments, and curves indicated the best local fit for the data are used to calculate the KD values by using a 1:1 binding model or steady state affinity.
(PDF)

**S13 Fig. The neutralization breadth of the polymerized mi3-1C4 against SARS-CoV-2 D614G strain and Omicron variants.** (A) The neutralization activities of mi3-1C4 (dark green line) and 1C4-SpyTag (light green line) against LV-based pseudoviruses of the SARS-CoV-2 D614G strain and Omicron variants. Data were collected from three technical replicates and displayed as means ± SD. The curves were analyzed by nonlinear regression (four-parameter) using GraphPad Prism (version 8.0.1). (B) A summary of the IC50 values of mi3-1C4 and 1C4-SpyTag calculated from the broad-spectrum neutralization assay in (A).
(PDF)

**S1 Table. Cryo-EM data collection, refinement, and validation statistics of the immune complexes.**
(PDF)

## Acknowledgments

We thank the members of the National Institute of Diagnostics and Vaccine Development for the technical assistance, especially Shuangquan Gao, Shaoyong Li and Feibo song.

## Author contributions

**Conceptualization:** Shaowei Li, Ying Gu, Peijun Zhang, Ningshao Xia, Qingbing Zheng.

**Data curation:** Yanan Jiang, Miaolin Lan, Ming Zhou, Gangshun Yi, Juan Shen, Liqin Liu.

**Formal analysis:** Ming Zhou, Juan Shen.

**Funding acquisition:** Peijun Zhang, Ningshao Xia, Qingbing Zheng.

**Investigation:** Miaolin Lan, Ming Zhou, Gangshun Yi, Juan Shen, Tingting Deng, Yang Huang, Yu Li, Yanling Lin, Zhenqin Chen.

**Methodology:** Hui Sun, Yanan Jiang, Miaolin Lan, Ming Zhou, Gangshun Yi, Juan Shen, Tingting Deng, Liqin Liu, Yang Huang, Yu Li, Jinfu Su, Yanling Lin, Zhenqin Chen, Lunzhi Yuan.

**Resources:** Gangshun Yi, Juan Shen.

**Supervision:** Lizhi Zhou, Tingting Li, Hai Yu, Tong Cheng, Yali Zhang, Lunzhi Yuan, Shaowei Li, Ying Gu, Peijun Zhang, Ningshao Xia, Qingbing Zheng.

**Validation:** Hui Sun, Yanan Jiang, Miaolin Lan, Ming Zhou, Gangshun Yi, Juan Shen, Liqin Liu, Jinfu Su.

**Visualization:** Hui Sun, Yanan Jiang, Ming Zhou, Gangshun Yi, Juan Shen, Tingting Deng, Liqin Liu.

**Writing – original draft:** Hui Sun, Yanan Jiang.

**Writing – review & editing:** Shaowei Li, Ying Gu, Qingbing Zheng.

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
