## [Decision Letter · Decision Letter 0]

27 Apr 2025

Structural-based Engineering of Neutralizing Antibody to Overcome SARS-CoV-2 Omicron Variants Immune Evasion

PLOS Pathogens

Dear Dr. Zheng,

Thank you for submitting your manuscript to PLOS Pathogens. After careful consideration, we feel that it has merit but does not fully meet PLOS Pathogens's publication criteria as it currently stands. Therefore, we invite you to submit a revised version of the manuscript that addresses the points raised during the review process.

Please submit your revised manuscript within 60 days Jun 23 2025 11:59PM. If you will need more time than this to complete your revisions, please reply to this message or contact the journal office at plospathogens@plos.org. Please include the following items when submitting your revised manuscript:

We look forward to receiving your revised manuscript.

Kind regards,

Tongqing Zhou, Ph.D.

Guest Editor

PLOS Pathogens

Sonja Best

Section Editor

PLOS Pathogens

Editor-in-Chief

PLOS Pathogens

Editor-in-Chief

PLOS Pathogens

orcid.org/0000-0002-7699-2064

**Journal Requirements:**

Please also provide the accession number for this dataset “EMD-xxx (WT-S:8H12:1C4:3E2).”

**Reviewers' Comments:**

Reviewer's Responses to Questions

**Part I - Summary**

Reviewer #1: Sun et al. investigated the reduced efficacy of the broadly neutralizing antibody 1C4 against SARS-CoV-2 Omicron variants. They clarified the antibody’s recognition mechanism and the molecular basis for its diminished neutralization activity against highly immune-evasive variants such as BA.5 and XBB. By utilizing the mi3-Spycatcher nanoparticle platform to achieve multivalent polymerization of 1C4, the study significantly improved its binding avidity and successfully restored its neutralizing capacity. By integrating structural biology, cellular functional assays, and evaluations in animal models, this work proposes a viable strategy for the engineering and optimization of broadly neutralizing antibodies. This work is meaningful and offers valuable insights into addressing the ongoing evolution of SARS-CoV-2. However, the following issues still need to be addressed.

Reviewer #2: This manuscript presents a timely and comprehensive investigation into the mechanisms of immune evasion by emerging SARS-CoV-2 Omicron subvariants and introduces an innovative antibody engineering strategy to counteract this challenge. Focusing on the broadly neutralizing antibody (bnAb) 1C4, the authors explore its reduced neutralization potency against recent variants such as BQ.1.1 and XBB—primarily due to the R346T mutation—and provide structural insights using cryo-electron microscopy. Despite a loss in neutralizing activity, 1C4 retains weakened spike binding and exhibits diminished ability to block ACE2 engagement. To overcome this, the authors employ a multivalent nanoparticle display strategy (mi3-1C4), which restores neutralization through enhanced avidity and demonstrates protective efficacy in a hamster model. This multidisciplinary approach, combining structural biology, binding assays, and in vivo studies, underscores the translational potential of multivalent antibody formats in the face of rapid viral evolution.

The study offers a strong conceptual foundation with clear relevance and scientific value. The idea that diminished antibody function can be rescued through avidity-driven engineering is both innovative and impactful. While the structural data and mechanistic insight are not yet fully integrated with the antibody engineering strategy, the study nonetheless advances the understanding of how bnAbs can be adapted to cope with emerging escape variants. The demonstration of functional rescue in both pseudovirus and live virus contexts is particularly compelling and reinforces the potential utility of this approach for therapeutic development.

That said, the manuscript requires major revisions to fully realize its potential. The presentation is currently hampered by significant language and clarity issues, as well as inconsistencies in figure labeling and data interpretation. Key methodological details are missing, and the proposed “lockdown” mechanism lacks definition and internal consistency. The inclusion of additional antibodies in the cryo-EM studies without accompanying functional data adds unnecessary complexity and may dilute mechanistic clarity. Moreover, statistical analysis is needed to strengthen the conclusions drawn from the in vivo data. The title should also be revised to more accurately reflect the scope and evidence presented in the manuscript.

In summary, this is a promising and high-impact study with clear relevance to the field of antibody therapeutics. While several aspects of execution and presentation need significant improvement, the underlying concept is strong and the data are directionally supportive of a meaningful advancement. With careful revision, improved structural interpretation, and clearer mechanistic framing, this work has the potential to make a valuable contribution to the ongoing effort to counteract SARS-CoV-2 immune escape. I encourage consideration for publication after major revision.

Reviewer #3: In this manuscript, Sun et al. report the engineering of a neutralizing antibody to overcome immune evasion by Omicron variants. Through neutralization, binding assays and cryoEM analysis, the authors identify the R346T in recent variants as a key residue to 1C4 escape. Additionally, the RBD-up, -down confirmations observed in different variants appear to contribute to immune evasion. Displaying 1C4 on a nanoparticle promotes efficient spike aggregation and viral cross-linking through multivalent binding, thereby enhancing neutralization activity and providing protection against Beta and XBB challenge in a hamster model.

Overall, the paper is well-constructed, the study aim is clear, and the results are of interest to the field. However, some methods, figure legends lack sufficient detail, and some results do not fully support the conclusions (see specific comments below). In my view, this manuscript can be published after these issues are addressed.

Reviewer #4: This paper examines why antibodies lose neutralizing potency against SARS-CoV-2 variants, and how to improve upon them. They solve many cryoEM structures of neutralizing antibodies bound to spike proteins from various variants, and show that one antibody in particular (1C4) may lose ability to neutralize XBB variants because it can no longer “lock” the RBDs in the down confirmation. They go on to multimerize IC4 using a spy tag system, which increases the neutralization against XBB and beta variants in mice and has significantly better potency in vitro. They then investigate the mechanism of action of this multimerized form of 1C4 and suggest that it may be different from un-multimerized IgG. While 1C4 presented as an IgG “locks” the RBD down, multimerized 1C4 may instead cause protein/virus aggregation, thus preventing infection. This is supported by negative stain EM images that show that the multimerized antibody causes aggregation of spikes and CryoEM images of SARS-Cov02 virion that show they may aggregate together more in the presence of multimerized antibodies.

**Part II – Major Issues: Key Experiments Required for Acceptance**

Reviewer #1: 1. In lines 161-168, although the N343 glycan appears to interact with 1C4 in the structural model, its direct involvement in antibody binding has not been confirmed. The authors are encouraged to perform glycan mutation or enzymatic deglycosylation experiments to determine whether the glycan is part of the epitope or indirectly contributes via conformational stabilization.

2. In lines 175-180, the author analyzed the impact of the R346T mutation on antigen–antibody binding through structural modeling, but the explanation remains insufficient. In addition to the potential loss of the hydrogen bond between R346 and Y50 due to the mutation, is there a cation-π interaction between R346 and W102? If so, is this interaction disrupted by the R346T mutation? It is recommended that the author further investigate the impact of the R346 mutation in the structural analysis.

3. Due to the influence of molecular flexibility, affinity estimation based on cryo-EM density intensity is not sufficiently reliable. Conclusions regarding antibody binding strength should primarily be supported by functional assays, such as surface plasmon resonance (SPR) measurements.

4. The study notes that 1C4 locks WT spike in a “three down RBD” conformation, whereas the BA.5 spike remains one RBD “up”. It would strengthen the manuscript to discuss possible mechanisms for this increased RBD-up propensity in BA.5, such as trimer destabilization caused by specific mutations.

5. Despite a measurable binding affinity between 1C4 and the XBB RBD, a complete loss of neutralization was observed. The underlying structural basis for this discrepancy should be further discussed.

6. This study lacks a control group involving the use of mi3-Spycatcher alone to rule out potential nonspecific effects of the nanoparticle platform. It is recommended that such a control be included, or that this limitation be explicitly addressed in the Discussion section

Reviewer #2: 1. The current title, “Structural-based engineering of neutralizing antibody,” is misleading, as there is no demonstrated connection between the structural findings and the antibody engineering strategy described. The manuscript does not establish how structural information guided the engineering of the mi3-1C4 construct, making the title inaccurate and the claim unsupported.

2. The inclusion of two additional antibodies, 3E2 and 8H12, in the cryo-EM analysis is not justified. These antibodies are not evaluated in functional assays such as neutralization or animal models. If the authors intended to test synergy, it is not demonstrated. Otherwise, their presence in the structural study may confound interpretation and introduce artifacts that obscure the mechanism of 1C4 alone.

3. Figure 4, which presents the animal study data, lacks any statistical analysis. Although the authors suggest that multivalent 1C4 provides enhanced protection, these claims are unsupported without statistical validation of viral load differences or protective efficacy. Including appropriate statistical tests is necessary to substantiate the conclusions drawn from these experiments.

Reviewer #3: 1. Fig 1A and 1B, Since the BA.4 and BA.5 spike sequences are identical, it is unnecessary to show both. It would be helpful to include more detailed information about the amino acid substitutions in the spike proteins in the “Materials and methods” section. Additionally, to help orient readers to the significance of R346T, the authors should note that the only difference between BQ.1 and BQ.1.1, or BA.2.75 and BA.2.75.2 (this data would be valuable to include but is currently mission), is the R346T substitution. XBB contains both the R346T and V445P substitutions, which may collectively contribute to the reduced binding/neutralization by 1C4.

2. Figure 3A: The authors should first demonstrate the binding of the three mAbs (preferably Fabs) to the variant spike proteins using the same method, prior to evaluating their ability to inhibit ACE2 binding. If the mAb and ACE2 are mixed before being added to the spike-coated plate, their binding will largely depend on their relative affinities. It is expected that mAbs with weak binding affinity for the spike will not effectively compete with ACE2. Given that the binding footprint of 1C4 only marginally overlaps with the ACE2 binding site, it is plausible that 1C4 IgG could partially interfere with ACE2 binding through steric hindrance.

3. Since several platforms aimed at enhancing antibody avidity and thereby increasing binding and/or neutralization have already been published, the authors should discuss the advantages and disadvantages and potential applications of their engineered antibody in this context.

Reviewer #4: One of the major claims in this manuscript is that multimerizing antibodies may decrease the frequency of which they need to be replaced in the clinic and “keep pace with viral evolution” but the data shown here indicates that strategy may only be practical for a small subset of antibodies. The group demonstrates the likely method of neutralization of 1C4 is “locking” of the RBD in the down conformation, and that 1C4 can still bind modern variants, but cannot lock the RBD. Surely, for antibodies that completely lose affinity to spike, multimerizing would have very little effect. This, combined with the fact that the increase (over regular IgG) in protection shown here conveyed by multimerizing 1C4 are marginal, severely diminished my enthusiasm for this manuscript. Further, it is extremely well established as a physical law of chemistry that multimerizing a protein will increase its apparent affinity (avidity effects).

There is some interesting information about the antibody 1C4 losing the ability to “lock” the RBDs in the down state over time. There is a wealth of structural data reported here, but the molecular/evolutionary mechanisms for how SARS-CoV-2 evades locking are not presented.

This may be because the resolution for the 1C4 WT spike data set is very low, based off very few particles with very few images taken, and there is no model built. Maybe this is the reason that no RBDs in the up conformation were observed? Perhaps a larger data set would reveal more classes with RBDs in the up conformation? Given that a large portion of the paper is based on the observation that 1C4 can lock the RBD, I would expect this data set to be much higher quality and have a model built to understand the molecular mechanisms. I would at a minimum expect multiple lines of evidence to be presented that support this method of neutralization.

**Part III – Minor Issues: Editorial and Data Presentation Modifications**

Reviewer #1: 1. In Figure 1G, the hydrogen bonding interactions involving the backbone are shown, and the participating backbone atoms should also be clearly indicated in the figure.

2. In the Results section, detailed descriptions of experimental procedures should be minimized. It is recommended that non-essential methodological content be moved to the Materials and methods section.

Reviewer #2: 1. The manuscript contains widespread typographical and grammatical errors that interfere with readability and clarity. Examples can be found on lines 77, 125, 301, and 367, among others. A thorough proofreading and overall improvement in English usage is necessary before the manuscript can be properly evaluated.

2. Figure legends are inadequately labeled and in some cases contain direct errors. For instance, lines 171–173 mention residues “K440” or “K44” in figure 1G, which are not visible or labeled in the figure. Additionally, in figure 2A, it is unclear which variant the antigen is derived from—this is critical information that should be clearly stated. Such omissions make it difficult to interpret the data and reduce the reliability of the figures.

3. Potential immunogenicity issue of using NP-based therapy is not discussed.

4. Structural modeling described in figure 1I and discussed in lines 175–183 lacks any methodological description. The manuscript should clearly state what modeling tools or approaches were used, how the models were generated, and how they were validated. This is essential for reproducibility and to assess the reliability of the conclusions drawn from these models.

5. In figure S1, the binding kinetics of XBB-RBD display a sharp Kon curve and an unusually high association rate. The authors do not provide any explanation for this atypical kinetic profile. Given the importance of binding affinity in the study’s conclusions, this observation needs to be addressed and interpreted.

6. The cryo-EM structure of the 1C4-spike complex was already published by the authors in reference 20. The manuscript does not explain what new information the current structures provide, or why repeating the structural analysis was necessary. This lack of justification weakens the rationale for including these new structural datasets.

7. The manuscript provides epitope-level detail for antibody binding, but does not adequately describe or analyze the paratope, i.e., the regions of 1C4 that engage the spike. Understanding the paratope is crucial for dissecting binding specificity and for any claims related to antibody engineering or escape mutations, and this should be addressed more thoroughly.

8. The cryo-EM structures in figure 2 are not discussed in sufficient detail. Their inclusion seems to serve only as a visual confirmation that the antibodies bind to the spike, without offering meaningful structural insight or analysis. If they are not contributing to the mechanistic understanding, the necessity of including them should be reconsidered or their interpretation expanded.

9. The manuscript proposes a “lockdown” mechanism of neutralization by 1C4, but this concept is inconsistently presented. It is unclear why the mechanism would be effective for some variants but not others, especially given that all tested variants are neutralized. The authors need to clearly define what the lockdown mechanism entails and reconcile it with their binding and neutralization data.

Reviewer #3: 4. Fig2C, it is not clearly indicated which variant- BA.2, BA.5, or XBB-the 3 mAbs are escaping from. Based on the text (line 218), it appears that all 3 mAbs escape from XBB. This should be clarified in the figure or legend.

5. Fig 2E, There are labeling errors in the figure. 1C4-VL is incorrectly labeled as 8H12-VH, and 3E2-VH and 3E2-VL are mislabeled as 8H12-VH and 8H12-VL, respectively. Please correct these to avoid confusion.

6. Figure 3F–H: In the legend, the phrase “Comparison of the ACE2 binding potency for the RBD in WT- and BA.5-spike complexed with 1C4” is misleading. The term “potency” does not accurately describe what Figures 3F–H are showing.

7. Fig 4C, An excess amount of 1C4-spytag should be used to ensure saturation of the mi3 binding site. The authors should clarify that the mi3-1C4 used in Fig 4F and in the animal studies are the same as that shown in Fig 4C, meaning that not all binding sites may be occupied by 1C4. Discussion of potential lot-to-lot variation would also be helpful.

8. Fig 5E, 5G, the y-axis label “viral RNA at copies/ml” is confusing and should be changed to copies/gram of tissues, and the full names of the organs should be used in the figures and/or legends for clarity.

8. Line 377: The phrase “significantly higher” appears to be incorrect in context. Based on the values provided, the mi3-1C4-treated hamsters showed lower viral RNA levels than the untreated group. Please revise accordingly.

9. Fig 6F, Please clearly describe how the samples were transferred from BSL-3 to BSL-2 in Fig 6F, specifically noting that the samples were fixed prior to transfer.

10. Line 655, “For the solid organ samples, we collect 1mg turbinate, 0.1 mg trachea and 0.1 mg lung in 1mL PBS”, did you mean grams instead of mg?

Reviewer #4: Line 135- typo

Line 205- I do not think we can infer anything about affinity from cryoEM data sets. Affinity should refer directly to biophysical experiments that measure affinity. FAB dentistry in cryoEM can result from many factors and does not necessarily relate to affinity.

Line 216- I am not sure that “diverse amounts of Fabs on different spikes is really relevant to the mechanism of action of these ABs. For example, differences in spike could cause slightly different FAB orientations which affects how many FABs can bind at the same time.

Figure 2a- does not really seem to be explained in the text or figures. It is unclear to me why 3E2 was injected one separately from the other in the competition assay3

Line 234- typo

Line 264- no evidence is given for why 1C4 locks some spikes but not others in the down state. Since there is a wealth of structural data presented, a structural mechanism for this should be delineated.

Line 456- “been proved to render the cross linking of trimeric spikes” does not quite make sense

Line 458- is capable to block should be “is capable of blocking”

Line 464-5 are very hard to understand what is being conveyed

Line 467- should be “…of virus evasion”

PLOS authors have the option to publish the peer review history of their article (what does this mean? ). If published, this will include your full peer review and any attached files.

**Do you want your identity to be public for this peer review?** For information about this choice, including consent withdrawal, please see our Privacy Policy .

Reviewer #1: **Yes: ** Xiangxi Wang

Reviewer #2: No

Reviewer #3: No

Reviewer #4: No

**Figure resubmission:**

**Reproducibility:**



---

## [Decision Letter · Decision Letter 1]

10 Sep 2025

PPATHOGENS-D-25-00661R1

Engineering a Multivalent Antibody Nanoparticle to Overcome SARS-CoV-2 Omicron Immune Evasion

PLOS Pathogens

Dear Dr. Zheng,

Thank you for submitting your manuscript to PLOS Pathogens. After careful consideration, we feel that it has merit but does not fully meet PLOS Pathogens's publication criteria as it currently stands. Therefore, we invite you to submit a revised version of the manuscript that addresses the points raised during the review process.

Please submit your revised manuscript within 30 days Nov 09 2025 11:59PM. If you will need more time than this to complete your revisions, please reply to this message or contact the journal office at plospathogens@plos.org. Please include the following items when submitting your revised manuscript:

We look forward to receiving your revised manuscript.

Kind regards,

Tongqing Zhou, Ph.D.

Guest Editor

PLOS Pathogens

Sonja Best

Section Editor

PLOS Pathogens

Sumita Bhaduri-McIntosh

Editor-in-Chief

PLOS Pathogens

orcid.org/0000-0003-2946-9497

Michael Malim

Editor-in-Chief

PLOS Pathogens

orcid.org/0000-0002-7699-2064

**Additional Editor Comments:**

Reviewer #2:

Reviewer #4:

**Journal Requirements:**

At this stage, the following Authors/Authors require contributions: Jinfu Su. Please ensure that the full contributions of each author are acknowledged in the "Add/Edit/Remove Authors" section of our submission form.

**Reviewers' Comments:**

Reviewer's Responses to Questions

**Part I - Summary**

Reviewer #2: The authors have addressed most of the previous concerns; however, several issues remain unresolved or insufficiently addressed. Below are the major and minor points that still require attention.

Reviewer #4: The authors have sufficiently addressed all my concerns, and I have no further comments.

**Part II – Major Issues: Key Experiments Required for Acceptance**

Reviewer #2: Major Comments

Comment 1: Addressed.

Comment 2:

In the rebuttal, the authors explain that the inclusion of two additional antibodies in the structural analysis was intended to “address a fundamental question: How do distinct antibody epitopes differentially respond to viral evolution.” This is an important and insightful rationale.

However, this motivation and its implications are not reflected in the revised manuscript. The Results and Discussion sections should explicitly elaborate on how the presented structures contribute to answering this fundamental question. Please integrate a clear discussion linking the structural data to this overarching objective.

Comment 3: Addressed.

Reviewer #4: (No Response)

**Part III – Minor Issues: Editorial and Data Presentation Modifications**

Reviewer #2: Minor Comments

Comment 1:

The authors stated that the manuscript was “carefully” and “thoroughly proofread,” yet most errors remain unchanged from the initial version, despite being specifically highlighted in the previous review.

Examples of uncorrected errors:

Line 122 (previously line 125): “…may does not lead to…”

Line 316 (previously line 301): “spacater”

In addition, numerous new errors were introduced during the revision process:

Lines 166–168: Ten residues are listed, but only nine are highlighted as the 1C4 epitope (green) in Figure S6. Residue T345 is missing.

Lines 168–179: The revision removed K440, but it is now unclear which of the remaining residues mediate the salt bridge interaction.

Lines 170–173: These lines describe the wild-type sequence but cite Figure 1G, which, according to the legend, corresponds to BA.5 RBD.

Lines 171–173: The text describes N440K and cites Figure 1G, yet neither K440 nor N440 is shown or labeled in the figure.

These issues indicate that the manuscript requires another round of careful proofreading and verification to ensure consistency between text and figures.

Comment 2:

The authors claim this point has been addressed, but no revision is evident.

Comment 3: Addressed.

Comment 4: Addressed.

Comment 5:

The authors cite Liu et al., Science 2021, 372(6541):525–530 to justify their observation of an atypical kinetic profile. However, there are several issues:

The cited paper does not include any author with the first or last name “Liu.”

The cited paper used a bi-valent analyte (see Figure 1B, top two curves) and applied a 1:2 curve fitting model (Table S1).

If the current study also used a bi-valent analyte, this must be explicitly stated, with clarification whether a 1:1 or 1:2 fitting model was applied in this manuscript.

The cited paper reports a maximum kon of 10^5 and koff of 10^-2, whereas this manuscript reports values of 10^8 and 10^2, respectively. Such a large discrepancy strongly suggests either an experimental artifact or an unusual biological finding. This issue needs to be discussed explicitly in the text.

Comment 6: Addressed.

Comment 7: Addressed.

Comment 8:

The revised manuscript now justifies including the cryo-EM structure in Figure 2, noting in lines 233–236:

“These structural findings… quantitatively reveal that many Omicron subvariants retain low-level antibody engagement.”

However, it remains unclear how this “quantitative” assessment was conducted.

Please provide a clear explanation of the methodology used to determine this quantitative measurement, or include supporting data to substantiate the claim.

Comment 9:

In the rebuttal letter, the authors state:

“Neutralization of Omicron subvariants by 1C4 likely involves alternative mechanisms, such as inhibition of membrane fusion or prevention of S1 shedding.”

If the authors believe that 1C4 neutralizes different variants through distinct mechanisms, this hypothesis should be explicitly stated in the manuscript itself. The Discussion should be revised to incorporate this point, rather than leaving it only in the rebuttal letter.

Reviewer #4: (No Response)

PLOS authors have the option to publish the peer review history of their article (what does this mean? ). If published, this will include your full peer review and any attached files.

**Do you want your identity to be public for this peer review?** For information about this choice, including consent withdrawal, please see our Privacy Policy .

Reviewer #2: No

Reviewer #4: No

**Figure resubmission:**

**Reproducibility:**



---

## [Editor Report · Decision Letter 2]

19 Nov 2025

Dear Dr. Zheng,

We are pleased to inform you that your manuscript 'Engineering a Multivalent Antibody Nanoparticle to Overcome SARS-CoV-2 Omicron Immune Evasion' has been provisionally accepted for publication in PLOS Pathogens.

Best regards,

Sonja M. Best, Ph.D.

Section Editor

PLOS Pathogens

Sonja Best

Section Editor

PLOS Pathogens

Sumita Bhaduri-McIntosh

Editor-in-Chief

PLOS Pathogens

orcid.org/0000-0003-2946-9497

Michael Malim

Editor-in-Chief

PLOS Pathogens

orcid.org/0000-0002-7699-2064
---

## [Editor Report · Acceptance letter]

Dear Dr. Zheng,

We are delighted to inform you that your manuscript, " 

Engineering a Multivalent Antibody Nanoparticle to Overcome SARS-CoV-2 Omicron Immune Evasion," has been formally accepted for publication in PLOS Pathogens.

Best regards,

Sumita Bhaduri-McIntosh

Editor-in-Chief

PLOS Pathogens

orcid.org/0000-0003-2946-9497

Michael Malim

Editor-in-Chief

PLOS Pathogens

orcid.org/0000-0002-7699-2064